



# Ice injected up to the Tropopause by Deep Convection: 1) in the Austral Convective Tropics

Iris-Amata DION[1], Philippe RICAUD[1], Peter HAYNES[2], Fabien CARMINATI[3] and Thibaut DAUHUT[4]

[1] CRNM, Meteo-France - CNRS, Toulouse, 31057, France
[2] DAMTP, University of Cambridge, Cambridge, CB3 0WA, United Kingdom
[3] Met Office, Exeter Devon EX1 3PB, United Kingdom
[4] Laboratoire d'Aerologie, Toulouse, 31400, France

Correspondence to: Iris-Amata DION (iris.dion@meteo.fr)

**Abstract.** The impact of deep convection on the water budget (water vapor and ice) from the tropical Upper Troposphere
(UT, around 146 hPa) to the Tropopause Level (TL, around 100 hPa) is investigated. Ice water content (IWC) and water
vapour (WV) measured in the UT and the TL by the Microwave Limb Sounder (MLS, Version 4.2) are compared to the
precipitation (Prec) measured by the Tropical Rainfall Measurement Mission (TRMM, Version 007). The two datasets,
gridded within 2°×2° horizontal bins, have been analyzed during the austral convective season: December, January and
February (DJF) from 2004 to 2017. MLS observations are performed at 01:30 and 13:30 Local Solar Time whilst the Prec
dataset is constructed with a time resolution of 1 hour. The new contribution of this study is to provide a much more detailed
picture of the diurnal variation of ice than is provided by the very limited (2 per day) MLS observations. Firstly, we show
that IWC represents 70 and 50% of the total water in the tropical UT and TL, respectively and that Prec is spatially highly
correlated with IWC in the UT (Pearson linear coefficient $R$=0.7). We propose a method using Prec as a proxy of deep
convection bringing ice up into the TL, during the growing stage of the convection. We validate the method using ice
measurements from the Superconducting Submillimeter-Wave Limb-Emission Sounder (SMILES) during the period DJF
2009-2010. Next, the diurnal amount of IWC injected into the UT and the TL by deep convection is calculated by the
difference between the maximum and the minimum in the estimated diurnal cycle of IWC in these layers and over selected
convective zones. Six tropical highly convective zones have been chosen: South America, South Africa, Pacific Ocean,
Indian Ocean, and the Maritime Continent region, split into land (MariCont-L) and ocean (MariCont-O). IWC injection is
found to be 2.73 and 0.41 mg m$^{-3}$ over tropical land in the UT and TL, respectively, and 0.60 and 0.13 mg m$^{-3}$ over tropical
ocean in the UT and TL, respectively. The MariCont-L region has the greatest ice injection in both UT and TL (3.34 and
0.42-0.56 mg m$^{-3}$, respectively). The MariCont-O region has less ice injection than MariCont-L (0.91 mg m$^{-3}$ in the UT and
0.16-0.34 mg m$^{-3}$ in TL), but has the highest diurnal minimum value of IWC in the TL (0.34-0.37 mg m$^{-3}$) among all oceanic
zones.

# 1 Introduction

Water vapour (WV) is one of the main greenhouse gases and has an important impact on the climate. WV in the stratosphere
has a direct radiative impact on surface temperatures and it participates in the warming of the troposphere and the cooling of





the stratosphere (Solomon et al., 2010; Birner and Charlesworth, 2017). It also has a strong impact on stratospheric chemistry, especially on the ozone equilibrium (Stenke and Grewe et al., 2005) as it contributes to the ozone destruction in

the polar regions via the formation of polar stratospheric clouds (Fahey et al.,1989; Solomon, 1999; Vogel et al., 2011). According to observations, stratospheric WV concentrations increased by ~30%  between 1980 and 2000 (Scherer et al, 2008; Hurst et al. 2011) and would increase by up to a factor of 2 over the twenty-first century (Gettelman et al., 2010; Maycock et al., 2013; Hegglin et al., 2014). After 2000, some studies have related the stratospheric WV drop in the tropics with the decrease of the Cold Point tropopause (CPT) temperature (Randel et al., 2006). It is known that tropospheric air

masses and water enter the stratosphere mostly in the tropics (Fueglistaler et al., 2009). In the tropics, the minimum of temperature at the tropopause regulates the stratospheric WV. The tropical tropopause layer (TTL) is defined by the layer between the level of maximum convective outflow (10-12 km, ~200 hPa) that closely corresponds to a minimum in ozone and the CPT temperature (16-17 km, ~100 hPa) (Gettelman and Forster, 2002;  Mehta et al., 2008; Birner and Charlesworth, 2017). Among this layer, in the present study, we separate and define the Tropopause Level (TL, ~17 km, ~100 hPa) to the

Upper Troposphere level (UT, ~14 km, ~146 hPa). In the tropics, the exchange between the troposphere and the stratosphere is larger over the region called Maritime Continent (MariCont), the region between the Indian Ocean and the West Pacific (Newell and Gould-Stewart, 1981), than elsewhere on earth. Newell and Gould-Stewart (1981) have called this strong exchange over the MariCont region, the 'stratospheric fountain'. In the tropics, transport from troposphere to stratosphere (TST) occurs through a combination of large-scale three-dimensional circulation and deep convection (e.g. Fueglistaler et al.,

2009; Randel and Jensen, 2013). The relative importance of the two processes continues to be debated (e.g. Alcala and Dessler, 2002; Liu and Zipser, 2005; Carminati et al., 2014).

Water is one of the rare chemical compounds to be present in 3 phases (solid, liquid and gas) in the atmosphere. However, in the TTL, because of the very low temperature, water is only found as vapor and ice.

In the tropics, the Brewer Dobson Circulation (BDC) leads the variations of WV at the tropopause (Potter and Holton, 1995)

to propagate upward in the tropical stratosphere, a phenomenon called the "Tape Recorder effect" (Mote et al., 1996). The globally average TTL and stratospheric WV is controlled by the TTL very low temperatures which regulate the lowest saturation mixing ratio that the air masses encounter (Hartmann et al., 2001; Fueglistaler et al., 2009; Khaykin et al., 2013) and impacts the water phase change. The amount of water injected into the UT-LS is still not well quantified. Sherwood and Dessler (2000) present a hypothesis on the stratospheric hydration and dehydration as two different processes which occur on

different time scales and involving vertical and horizontal air mixing into the TL. Ueyama et al. (2015) have shown that tropical waves dehydrate the TL (at 100 hPa level) by 0.5 ppmv while the cloud microphysical processes and convection moisten the TL by 0.7 and 0.3 ppmv, respectively. Some authors have shown that fast overshoots from deep convection transports very rapidly water upward (from few minutes to one hour) into the stratosphere (Pommereau et al., 2010; Dauhut et al., 2015). Liu and Zipser (2005) have found that 1.3% of tropical convection systems reach 14 km and 0.1% of them may

even penetrate the 380 K potential temperature level which corresponds to the CPT. Dauhut et al. (2015) have estimated that 18% of the water mass flux across the 380 K potential temperature level is due to overshooting deep convection. Avery et al.



(2017) have also suggested that tropical convective ice cloud and associated cirrus sublimating at unusually high altitudes might also have a role in stratospheric hydration. The other path for water is the slow vertical advection (300 m.month−1) governed by radiative heating (Holton and Gettelman, 2001; Gettelman and Forster, 2002; Corti et al., 2006; Fueglistaler et al., 2009). Once into the UT or the TL, the amount of the relative humidity of the layer impacts the ice nucleation, growth and sedimentation (Peter et al., 2006). The amount of water in solid phase brought by deep convection into the UT-LS is not well quantified and the associated processes are not well understood. Thus, the aim of the present study is to quantify the role of the deep convection on the hydration and dehydration of the TL. For that, in this study, we focus on the diurnal injection of ice in the UT and the TL from the deep convective activity.

The diurnal cycle is one of the most fundamental modes of variability of the global climate system and is associated with large and well-defined variations in the solar forcing (Yang and Slingo, 2001). The amplitude of the surface thermal diurnal cycles in the tropics is ten times stronger than the amplitude of the surface thermal annual cycle (Beucher, 2010) and the diurnal cycle of deep convection is mainly governed by the surface thermal diurnal cycle. Understanding the diurnal cycle of the total water in the tropical UT, TL and LS contributes to characterize the total water exchange between the UT and the LS in the tropics. Many studies (e.g. Jiang et al., 2014) have shown that the diurnal cycle of ice in the UT is linked to the diurnal cycle of the convection. The maximum of deep convection in the UT has been shown to peak in the local afternoon over land and in the local early morning over ocean (Liu and Zipser, 2005). Based on these studies and with data from the space-borne Microwave Limb Sounder (MLS) instrument, Carminati et al. (2014) studied the diurnal cycle of WV and ice in the TL over Africa and South America and have shown the presence of a strong diurnal signal over the two continents. Suneeth et al. (2017) have shown the differences in the diurnal cycles of CPT altitudes and CPT temperatures over tropical land and ocean. The diurnal cycle of temperature has also been studied in the TL (Khaykin et al., 2013) showing that the amplitude of temperature anomalies increases with the intensity of the convection.

In summary, the processes driving the total water diurnal cycle are different in the UT and at the TL, e.g. only the deepest convection directly impacts water content at the TL and saturation is more frequent at the TL than in the UT. Since the total water diurnal cycle in the TTL is still not well described in global models (Jiang et al., 2014), space-borne observations may improve our understanding of the processes occurring in this layer.

The present study addresses the question relative to the diurnal cycle of total water over tropical deep convective areas. Africa and South America are two land regions of intense deep convective activity in the tropics, together with the MariCont region (made of lands and oceans). However, the deep convective systems occur more frequently and strongly over the MariCont region (Liu and Zipser, 2015) than over the other continental zones. It is known that excess of energy received by tropical surfaces is mainly balanced and transferred over land by convection toward mid-latitudes and over ocean by both convection and oceanic circulation. However, over the MariCont region, seas and oceans currents transferring energy toward mid-latitudes are slowed down and blocked by the thousands of islands. Consequently, the excess of energy in the oceanic surface over the MariCont region is mainly balanced by transfer from convection. The diurnal cycle of WV and ice has already been studied over Africa and South America and over the whole tropical band (see e.g. Liu and Zipser, 2009; Millán



et al., 2013; Carminati et al., 2014). The MariCont region is the driest and coldest area at the CPT and the water budget in the TL over the MariCont region separating MariCont land and MariCont ocean is still unknown. Furthermore the physical mechanism governing the diurnal cycle of convection and surface precipitation over ocean is still an open question.

In order to understand the total water diurnal cycle in the TL, we first use 13 years of the ice water content (IWC) and WV measurements from MLS (Version 4.2x) and surface precipitations (Prec) from the TRMM instrument. Then, we use WV, IWC, total water ($\sum W = WV + IWC$), relative humidity with respect to ice (RHI) and temperature (TEMP) from MLS and Prec from TRMM over the period 2004 to 2017 in order to better understand the water fraction (IWC/$\sum W$) in the UT (~146 hPa) and the TL (~100 hPa, around the CPT). The relationships between surface precipitation and the processes in the UT and the TL are investigated. Our study intends to evaluate the mechanisms affecting the diurnal cycle of total water in the TL above land and ocean with a particular emphasis over the MariCont region due to its topographical complexity. Thus, the diurnal cycle of ice (provided by MLS but with very poor time resolution) is improved and better understood thanks to the relation with the diurnal cycle of precipitation (provided by TRMM with a much better time resolution).

The main datasets used are presented in Section 2. The methodology developed to establish the link between the diurnal cycle of surface processes and the diurnal cycle of water in the UT and TL is presented in Section 3. The estimated diurnal injection of IWC in the UT and TL from deep convective activity is presented in Section 4. Finally, the influence of the convective dissipating stage on the decreasing phase of the IWC diurnal cycle in the UT and the TL is discussed in Section 5 and conclusion will be drawn in Section 6. This paper contains many abbreviations and acronyms. To facilitate reading, we compile them in the Appendix.

## 2 Datasets

### 2.1 MLS

MLS is a microwave instrument aboard the NASA's Earth Observing System (EOS) Aura platform (Livesey et al., 2017), measuring in a limb viewing geometry to maximize signal intensities and vertical resolution. The MLS instrument is aboard a sun-synchronous near-polar orbiter completing 233 revolution cycles every 16 days giving a daily global coverage with about 14 orbits. Aura is crossing the equator at 01:30 local time (LT) and 13:30 LT. Among all the atmospheric parameters measured by MLS, we focus on: IWC, WV, TEMP and RHI. IWC (mg m$^{-3}$) is valid between 215 and 82 hPa, WV (ppmv) between 316 and 0.002 hPa, TEMP (K) between 261 and 0.001 hPa and RHI (%) between 1000-1.0 hPa. The MLS IWC sensitivity thresholds do not allow to detect low ice content such as cirrus outflow associated with convective events. Consequently, MLS IWC sensitivity will mostly be able to detect ice from convective cores. Following Livesey et al. (2017), we will not consider IWC measurements less than 0.02 mg m$^{-3}$.

The MLS v4.2x data processing validated by Livesey et al. (2017) presents significant and minor differences with the previous MLS version. The total random noise in v4.2x IWC is larger than the one in v2.2. Compared with the v3.3 of MLS



(used for instance in Carminati et al., 2014), the v4.2 improves IWC composition profiles in cloudy regions especially in the upper troposphere over the tropics and improves the WV at global scale (Livesey et al., 2017).

Level 2 data quality tests (filter and data screening) from the v4.2x have been applied to WV, IWC, RHI, and TEMP as suggested by Livesey et al. (2017). The MLS measurements estimated from the averaging kernels (Livesey et al., 2017) are used to characterize the distribution of the four parameters on two layers: one at 146 hPa (UT) and one at 100 hPa (TL). The vertical resolution of the IWC measurements is ~3 km, the horizontal resolution is ~300 km and 7 km along and across the track, respectively. The WV vertical resolution is 2.5 ± 1.2 km and the WV along track horizontal resolution is between 170

and 350 km. The TEMP vertical resolution is 4.5 km.

In the study of Carminati et al. (2014), a 10°×10° resolution was chosen with seven years of MLS data processed with version 3.3 (v.3.3). In our study, MLS data have been averaged over a much longer period (2004 to 2017) in a much higher horizontal resolution of 2°×2° (~200×200 km). In each 2°×2° bin, and for the 13 years average, any bin with less than 60 measurements in the daytime (Day at 13:30 LT) or the nighttime (Night at 01:30 LT) datasets is excluded in order to have

significant statistics. Thus, the maximum of measurements per bin during the 13 years period is ~ 470 and the minimum is fixed to be 60. The number of measurements over the MariCont region is on average the lowest over the tropics. Consistent with Carminati et al. (2014) and Liu and Zipser (2005), we use the difference between the MLS measurements performed during the Day and the Night to study the "Day minus Night" (Day−Night) signal.

## 2.2 TRMM

The TRMM satellite was launched in November 1997 (https://trmm.gsfc.nasa.gov/publications_dir/publications.html). TRMM carried 5 instruments: a 3-sensor rainfall suite, the precipitation radar, the TRMM Microwave Imager, the Visible and Infrared Scanner and 2 related instruments, the Lightning Imaging Sensor and the Clouds and the Earth's Radiant Energy System. In our study, we use the Algorithm 3B42 producing the TRMM merged high quality to infrared precipitation and root-mean-square precipitation-error estimation with the TRMM version 007. The rainfall level 3 measurements (mm hr$^{-1}$)

are vertically integrated precipitation (Prec). TRMM Prec data are provided in the tropics within a 0.25°×0.25° horizontal resolution extending from 50°S to 50°N. TRMM datasets used in our study in coincidence with the MLS period of measurements from 2004 to 2017. In order to compare to the MLS measurement, we degraded the TRMM resolution to 2°×2° in averaging the TRMM measurements in each box of 2°x2° of MLS. Furthermore, as TRMM follows an orbit precession, shifting few minutes per day, the 24-hour diurnal cycle can be averaged over the period of study, with 1-hour

resolution over the 24 hour diurnal cycle.

## 3 Relationship between Prec and water budget in the austral convective UT and TL

In this section, we analyze the relationships between Prec and the water budget in the UT and the TL during the DJF season, highly convective season of the southern hemisphere. Whereas the water budget in the TL is mostly affected by deep convection, Prec is driven by both shallow and deep convection.



## 3.1 Tropical distribution

The daily averaged ((D+N)/2) Prec measured by TRMM at the resolution of 0.25°×0.25° and 2°×2° (the same horizontal resolution as MLS) are shown in Figures 1a and b, respectively. The fine structure of the horizontal distribution of Prec at the high resolution (0.25×0.25°) over South Africa, South America, Western Pacific Ocean and the Maritime Continent are degraded when considering Prec at low resolution (2°×2°). Figure 1c presents the Local Solar Time (evaluated for each bin of 0.25°×0.25°) at which the diurnal cycle of Prec reaches its maximum in TRMM observations showing the large variability and complexity of the diurnal maximum of Prec over the tropics. The daily averages of IWC, WV, IWC fraction (IWC/$\sum$W), TEMP and RHI from MLS in the tropics are shown in Figure 2 for the UT (at 146 hPa) and Figure 3 for the TL (100 hPa).

In DJF, local maxima of Prec, IWC, WV, IWC fraction and RHI in the UT (Fig. 2) are found over the main convective areas: South America, South Africa, MariCont, North of Australia, along the Inter-Tropical Convergence Zone (ITCZ) and the South Pacific Convergence Zone (SPCZ). Maximal values of Prec, IWC, WV and RHI over the whole tropics are located over the MariCont (and North Australia) (Figs. 1a and b and Figs. 2a, b, c and e). Minima of Prec, IWC, WV, IWC fraction and RHI are found over Eastern Pacific Ocean and South Atlantic Ocean. Over MariCont region, we observe the maximum of tropical IWC and the minimum of temperature (Figs. 2a and d). The IWC fraction shows that, over the region of high Prec (> 0.20 mm h$^{-1}$), up to 70% of the $\sum$W is composed of ice. The RHI is higher over land than over ocean but does not reach saturation (RHI < 100% at 2° resolution). In summary, there is a strong spatial link between Prec and water (IWC and WV) in the UT.

In the TL (Fig. 3), IWC is lower than in the UT (~3 mg m$^{-3}$ at 146 hPa vs < 1 mg m$^{-3}$ at 100 hPa) but the horizontal distribution of IWC is correlated with the horizontal distribution of Prec and shows maxima over South America, South Africa, the MariCont region and Western Pacific Ocean. The main difference between the UT and the TL is the minimum of WV observed over the MariCont region (> 8 ppmv at 146 hPa and < 3 ppmv at 100 hPa). According to the difference between the UT and the TL, WV decreases more with altitude over the MariCont region compared to all other tropical regions. Consistently, TEMP is lower at 100 hPa (near the CPT) than at 146 hPa and its value is the lowest over the MariCont region. The IWC fraction is larger over MariCont than elsewhere in the tropics in the TL ( near 78%). While WV decreases by more than 8 ppmv in the TL over the MariCont region compared to the UT, the RHI in the TL reaches high values (RHI ~100 %) highlighting a saturated environment over the central South America, Africa, MariCont and the Western Pacific Ocean.

To investigate the vertical distribution and the diurnal cycles of water species in the TL, we have defined seven tropical convective zones shown in Figure 4: South America (SouthAm, 0°N-30°S), South Africa (SouthAfr, 0°N-30°S), Pacific Ocean (PacOc, 0°N-30°S; 180°W-150°W), Indian Ocean (IndOc, 0°N-30°S; 60°E-90°E), MariCont (10°N-15°S; 90°E-160°E), MariCont land (MariCont-L) and MariCont ocean (MariCont-O). Land and ocean over the study zones have been separated using the Solar Radiation Data (SoDa, **http://www.soda-pro.com/web-services/altitude/srtm-in-a-tile**), providing a TIFF image with values from the Shuttle Radar Topography Mission (SRTM) Digital Elevation Model. Each



zone is defined at a horizontal resolution of 2°x2°.

**3.2 Water budget in the UT and the TL**

To investigate the processes in the UT and TL which drive the water budget, the vertical profiles of TEMP, WV, IWC and RHI are shown for the different study zones in Figure 5. The tropical CPT is found between 100 and 80 hPa (see Fig. 5a). The MariCont CPT is the coldest in the tropics (Fig. 5a). To complete comparison between study zones, the DJF average of the pressure level of the tropopause is represented over all the tropics in Figure 6 from the National Centers For Environmental Prediction (NCEP). While the East of the MariCont, East of SouthAfr and North of SouthAm present the highest tropopause (~105 hPa), PacOc is presenting the lowest tropopause (at ~120 hPa). Regarding WV, the tropical UT is more humid than the TL: the decrease from the UT at 146 hPa to the TL at 100 hPa is about 7 ppmv for WV, and 1 mg m$^{-3}$ for IWC (the two dashed lines in Figs. 5b and c, respectively). These results are consistent with previous studies all presented in Fueglistaler et al. (2009).

RHI is lower in the UT than at the TL by ~10 % (Figs. 2d and 3d, respectively). Since the UT is on average sub-saturated, convective lifted ice can sublimate and can be seen as a source of WV. At the TL, the atmosphere is close to saturation (RHI ~100%) over SouthAm, SouthAfr, MariCont and the Western Pacific Ocean (Fig. 3). The convective lifted ice does not sublimate as rapidly as in the UT and other processes may become significant as e.g. sedimentation or downdraft advection inside the convective system. According to Allison et al. (2013), the TL is the level of greatest dehydratation because of the supersaturation with respect to ice. With supersaturation condition, the excess of water vapour can condensate on existing ice crystals (or form new ones in presence of condensation nucleus) allowing the crystals to grow and sediment which dehydrates the layer. Conversely, hydration may occur in the LS (if reached by convection) because this layer is subsaturated with respect to ice (Allison et al., 2013).

**3.3 Space correlation between Prec and water in the UT**

In order to quantify the relationship between deep convection and the water budget injected in the UT, the correlation between both WV and IWC in the UT and Prec is analysed. We have calculated the Pearson linear correlation coefficient $R$ between the horizontal distribution of Prec and both IWC and WV at 146 hPa during the Day, the Night and the Day-Night over the study zones (see Table 1). We denote Prec, IWC and WV at 146 hPa during Day, Night and Day-Night, by $Prec_x$, $IWC_x^{146}$ and $WV_x^{146}$, respectively (where x= Day, Night or Day−Night). The spatial correlation between IWC and Prec is defined by the following equation:

$$IWC_x^{146} = \alpha_x \times Prec_x + \beta_x \qquad (1)$$

where $\alpha_x$ is the regression coefficient and $\beta_x$ *is* the offset.





The spatial correlation for the 6 zones is shown in Table 1. The spatial correlation of $IWC_{Day}^{146}$ and $IWC_{Night}^{146}$ with Prec is high ($R \sim 0.7$-$0.9$) over SouthAm, SouthAfr, PacOc and IndOc and smaller ($R \sim 0.5-0.6$) over the MariCont region (MariCont-L, MariCont-O and MariCont). The spatial inhomogeneity of the MariCont surface with thousands of islands, seas and coastal areas may probably explain the lower space correlation between the Prec and $IWC^{146}$. The regression coefficient between $Prec_x$ and $IWC_x$ (Table 1) $\alpha_x$, can be interpreted as the efficiency of the convective system to inject ice in the UT.

Comparing $\alpha_x$ in Day and Night conditions, SouthAfr, SouthAm, MariCont-L, MariCont-O and MariCont stand out with larger Day values ($\alpha_D = 7.5$-$15.0$ mg m$^{-3}$ mm$^{-1}$ h) than IndOc and PacOc or than Night values ($\alpha_x = 4.7$-$7.7$ mg m$^{-3}$ mm$^{-1}$ h). Over land, all regions under consideration show a better efficiency to inject ice in the UT during Day than during N. is $\beta_x$ the IWC background. Over land and ocean regions, $\beta_x$ is close to zero. $\beta_{Day}$ is the highest over the MariCont-L (1.1 mg m$^{-3}$) and null over SouthAm (0.0 mg m$^{-3}$). Over the oceans, there is no significant differences between $\beta_{Day}$ and $\beta_{Night}$ for the

injected ice in the UT, except over PacOc.

Following Liu and Zipser (2009) and Carminati et al. (2014), we also study the difference between Day and Night, to obtain information on the $IWC^{146}$ diurnal cycle. Figure 7 shows the $Prec_{Day-Night}$, $IWC_{Day-Night}^{146}$ and $WV_{Day-Night}^{146}$, respectively. Convective regions such as SouthAm, SouthAfr, even along the ITCZ, are characterized by a positive Day−Night signal in $Prec_{Day-Night}$ and $IWC_x^{146}$ with values greater than 0.10 mm hr$^{-1}$ and 0.20 mg m$^{-3}$, respectively (Figs. 7a and b). The spatial

correlation $R$ between $Prec_{Day-Night}$, and $IWC_{Day-Night}^{146}$ is greater over the 3 land zones ($R \geq 0.6$-$0.7$) than over the 3 ocean zones ($R \leq 0.1$-$0.5$). MariCont-O region shows the highest oceanic correlation ($R \sim 0.5$) compared to the PacOc and IndOc regions ($R = 0.2$ and 0.1, respectively). The low $R$ values over IndOc and PacOc could be explained by: i) the low $IWC_{Day-Night}$ variability compared to the high $Prec_{Day-Night}$ variability observed over ocean and ii) the low amplitude of the deep convection diurnal cycle over ocean (Liu and Zipser, 2005). We could hypothesize that $IWC^{146}$ over the MariCont-O

is more influenced by local deep convection than over IndOc and PacOc. However, considering $IWC^{146}$ and Prec during Day and Night independently, we find good correlation over the ocean ($\sim 0.6$-$0.8$ mg m$^{-3}$).

The spatial correlation $R$ between $Prec_x$ and $WV_x^{146}$ remains low in all configurations, with values of 0.-0.5 (Table 1). The spatial correlation between the $WV_{Day-Night}^{146}$ and $Prec_{Day-Night}$ shows an average over the 7 study zones of $\mu_{Zones} \sim 0.07$ with small negative correlations over SouthAm and SouthAfr and with a regression coefficient $\alpha_x$ between $WV^{146}$ and Prec

calculated to be $\alpha_x = 0$ (Table 1). We conclude that WV is not spatially correlated with the diurnal cycle of the deep convection. These results are consistent with results from Liu and Zipser (2009) and Carminati et al. (2014) showing that the Day−Night signal of WV measured by MLS is very different with the Day−Night signal of IWC measured by MLS in the UT over land.



Deep convection does not inject directly WV in the UT. Modelling studies from Cloud Resolving Models based on the
Hector convective system in North of Australia (Dauhut et al., 2015) show that WV in the UT and TL is produced by ice
sublimation. The onset of the WV increase is delayed by 1-3 hours with respect to the onset of deep convection (~12:00 LT).
To sum up, the space correlation $R$ between $IWC_{Day,Night}^{146}$ and $Prec_{Day, Night}$ ($R$ ~0.7) is larger than the space correlation $R$
between $WV_{Day, Night}^{146}$ and $Prec_{Day, Night}$ (R ~0.2). For that reason, we now focus on the diurnal cycle of ice and on the amount
of ice injected in the UT and the TL by deep convective systems traced by Prec.

**3.4 Diurnal cycle of Prec**

This section is presenting the diurnal cycle of Prec over the 6 study zones during the Southern Hemisphere convective
season (DJF) from 2004 to 2017 over land and ocean, respectively (Figures 8a and b).
Over land, the amplitude (peak to peak) of the Prec diurnal cycle varies between 0.2 and 0.3 mm h$^{-1}$, with simultaneous
minima between 08:00 and 09:00 LT within the regions. MariCont-L area shows the driest minimum (0.09 mm h$^{-1}$) and the
wettest and latest maximum (16:00-17:00 LT, 0.385 mm h$^{-1}$). SouthAm and SouthAfr maxima are found at 14:00-15:00 LT
and 15:00-16:00 LT, respectively.
Over ocean, the amplitude of the Prec diurnal cycle varies between 0.08 and 0.1 mm h$^{-1}$, with simultaneous minima between
17:00 and 18:00 LT. However, the maxima are spread between 01:00 and 06:00 LT.
The main difference between Prec diurnal cycle over tropical ocean and land are: i) morning maximum over ocean compared
to early afternoon over land, ii) peak to peak diurnal cycle of Prec is larger over land than over ocean and iii) minima can be
twice lower over land (0.1 mm h$^{-1}$) than over ocean. The comparison of the Prec diurnal cycle over MariCont and MariCont-
O or MariCont-L highlights the importance of analyzing separately the diurnal cycle over land and over ocean in this region,
because MariCont-O and MariCont-L have two distinct behaviours which are hidden when the diurnal cycle is only
considered over the MariCont global area.

**3.5 Time correlation between Prec and IWC during the growing convective phase in the UT**

In this section, we quantify the link between deep convection reaching the tropopause and Prec.The diurnal cycle of Prec is
compared to the diurnal cycle of the frequency of overshooting precipitation features (OPFs) events (i.e. the proportion of all
OPF events that occurs in various ranges of time of day, adapted from Figure 3 of Liu and Zipser (2005) from global satellite
radar measurements) over land (Figure 9a) and ocean (Figure 9b) in the tropical band (20°S-20°N). The OPFs considered by
Liu and Zipser (2005) are tropical deep convective systems penetrating the tropical TL (called $Z_{tropopause}$ in Liu and Zipser
(2005)). The OPFs are calculated from measurements from January 1998 to November 2000 and from December 2001 to
December 2003 based on TRMM precipitation radar (PR) measured reflectivity into precipitation features (PFs) and using
the method described by Nesbitt et al. (2000). The OPFs can be identified thanks to the high vertical resolution of the
TRMM PR.



Over land, the diurnal cycle of Prec and OPF shows a maximum at 16:00-17:00 LT and a minimum at 09:00-10:00 LT (Fig. 9a). The duration of the growing phase, defined as the period between the minimum and the maximum of Prec, is the same for OPF and Prec, and lasts $\Delta t = 7$ hours. The major difference in the diurnal cycles of Prec and OPF occurs after the

growing phase: the fraction of OPF decreases more rapidly than that of Prec and stops decreasing around 00:00 LT whereas that of Prec continues to decrease. This difference can be explained by the contribution to Prec of 1) the convection that does not reach the stratosphere and 2) the mature phase of the OPF that produces stratiform rain for some hours after the overshooting time. These results over land are consistent with Pereira et al. (2005) showing that the shallow convection is associated with 10% of precipitation over Amazonian region. Although OPF and Prec diurnal cycles are not calculated over

the same study period, we can deduce from the present analysis that during the growing phase, the time evolution of Prec is a good proxy for the time evolution of deep convection reaching the TL (OPFs).

Over ocean, the sea surface temperature (SST) does not have a strong diurnal cycle (less than ~1 K) with a peak in the early afternoon (e.g., Chen and Houze, 1997; Stuart-Menteth et al., 2003). The convective clouds usually develop after the SST peak, during the afternoon and during the night with a maximum in the early local morning. The diurnal cycle of Prec and

OPF (Fig. 9b) over the ocean have a maximum over night (05:00-06:00 LT) and a minimum in the end of the afternoon (18:00-19:00 LT for the OPF and 20:00-21:00 LT for the Prec). The diurnal cycles of OPF and Prec over the ocean show similar amplitudes in deviation from the mean. This result is consistent with Peirera et al. (2005) showing that the shallow convection is associated with only 3% of precipitation over Eastern Pacific region.

In summary, during the growing stage of the convective system, Prec is a good proxy of deep convection reaching the TL

over land and over the ocean, which will then be used to interpret the time evolution of IWC in the UT and the TL.

## 4. Amount of ice injected in the UT and the TL by deep convection

In this section, we present the method we have developed to estimate the diurnal cycle of IWC in the UT $\left( IWC^{UT}(t) \right)$ and in the TL $\left( IWC^{TL}(t) \right)$ based on the diurnal cycle of Prec and on the two per day IWC measurements by MLS. Like in the

previous section for OPF, we separate the diurnal cycle of IWC into a growing phase (period when deep convection develops to reach the UT and the TL), and a decreasing phase (period when deep convection dissipates). During the growing phase, Prec is a good indicator (proxy) of deep convection both over land and ocean (section 3.5). This is not true during the decreasing phase, when shallow convection can have a significant impact on Prec.Since IWC is spatially highly correlated with Prec and since the deep convection (OPFs) bringing ice into the TL increases with Prec during the growing phase, we

expect ice to be injected up to the UT and the TL with a delay $\left( \delta t^{UT,TL} \right)$ after the onset of the deep convection $t_{on}$ during the growing phase.

The method developed focuses on the growing stage of deep convection. We quantify the amount of ice injected to the UT and the TL and determine the onset time of the ice injection and its duration. The amount of ice injected will be calculated from the estimation of the amplitude of the diurnal cycle of IWC in the UT and the TL.





### 4.1 IWC diurnal cycle in the UT and TL


The method to estimate $IWC^B(t)$ $(B = UT\ or\ TL)$ is presented in this section. Figure 10 illustrates the methodology to find the $IWC^B(t)$ over land (the diagram over ocean would look different since maxima appear during local night $t_{on}$ ).is defined as the time when Prec starts to increase and $t_{on}^B$ is defined as the time when $IWC^B$ starts to increase. Our two main hypotheses are to assume that: i) $IWC^B$ starts to increase later than Prec (a delay $\delta t^B$ is assumed) because convective systems precipitate before reaching UT and TL, as follow:


$$t_{on}^B = t_{on} + \delta t^B \qquad (2)$$

and ii) the $IWC^B(t)$ is proportional to the $P(t)$ during the convective growing phase (see the period of proportionality in Fig. 10). This hypothesis considers deep convection represented by Prec as the main process bringing ice into the UT and the TL. Based on these two hypotheses and knowing $P_x$ and $IWC_x^B$ at $t_x$ and $t_{on}^B$, we estimate $IWC^B(t_{on}^B)$ as follows:


$$IWC^B(t_{on}^B) = \frac{Prec(t_{on}^B)}{Prec_x} \times IWC_x^B \qquad (3)$$

where x= $Day$ or $N$. We use $t_x = t_{Day}$ for land since only the Day measurement occurs during the growing phase over land, and $t_x = t_{Night}$ over sea since only the Night measurement occurs during the growing phase over ocean. Knowing $IWC^B(t_{on}^B)$ and $t_{on}^B$, the time evolution $IWC^B(t)$ during the growing phase is then:


$$IWC^B(t) = \begin{vmatrix} \dfrac{Prec(t)}{Prec(t_{on}^B)} \times IWC^B(t_{on}^B) & \text{if } Prec(t) > Prec(t_{on}^B) & (4) \\[2em] IWC^B(t_{on}^B) & \text{else.} & (5) \end{vmatrix}$$

Our method can then estimate the magnitude of the diurnal variation in the UT and the TL ($\Delta IWC^B$) over different regions of the tropics:

$$\Delta IWC^B = IWC^B(t_{end}) - IWC^B(t_{on}^B) = IWC_{max}^B - IWC_{min}^B \qquad (6)$$


with $t_{end}$ being the time of the diurnal maximum of Prec. $t_{end}$ can be considered as the time of the end for ice injection or the time from which the downdraft processes are more important than the updraft processes. Thus, we define $\delta t_{end}^{UT}$ as the delay between $t_{end}$ and $t_{end}^{UT}$. Following the hypothesis ii that considers that deep convection is the main process bringing ice into the UT and the TL, calculated $\Delta IWC^B$ represents the amount of ice injected by deep convection in these two layers. Thus, the duration of the injection of ice during the growing phase, $\Delta t^B$ is then:


$$\Delta t^B = t_{end} - t_{on}^B \qquad (7)$$



In order to validate this method, we compare the estimation of the amount of ice injected $\left(\Delta IWC^B\right)$ into the UT and the TL with the amount of ice measured in the troposphere and in the TL by the Superconducting Submillimeter-Wave Limb-Emission Sounder (SMILES) instrument on board the International Space Station (ISS) during the short convective period of December 2009 to February 2010.


## 4.2 Validation of the method with SMILES measurements

The SMILES instrument measured IWC between 120 and 80 hPa and the overall measurement of tropospheric ice, the partial Ice Water Path (pIWP) integrated between 1000 and 180 hPa, from October 2009 to April 2010. This period is an El Nino Southern Oscillation (ENSO) period, increasing the convective activity over South Africa and the Western Pacific

Ocean and decreasing the convective activity over the MariCont region and South America. The diurnal cycle of IWC and pIWP calculated from SMILES has shown the well separated signal over tropical land and ocean (Millán et al., 2013; Jiang et al., 2014). The anomalies of the diurnal cycle of IWC measured by SMILES near the TL, the diurnal cycle of pIWP in the troposphere and the diurnal cycle of Prec measured during the convective period of December 2009 to February 2010 are presented in Figure 11a over the three land study zones and in Figure 11b over the three ocean study zones, both with a

running average of 6 h. $t_{on}$, $t_{on}^{UT}$ and $t_{on}^{TL}$ are the times of the minimum of P ($Prec_{min}$), pIWP ($pIWP_{min}$), and IWC ($pIWP_{min}$), respectively.

Over land, during the growing phase, pIWP is proportional to Prec (Fig. 11a). Assuming that $IWC^{146}$ evolves like pIWC, this implies that $t_{on}^{UT} \approx t_{on}$ $\left(\delta t_{on}^{UT}=0\right)$ at ~ 08:00-09:00 LT, and $t_{end}^{UT} \approx t_{end} + \delta t_{end}^{UT}$ at ~ 16:00-17:00 LT, with $\delta t_{on}^{UT} \leq 1$ h and $\delta t_{end}^{UT} \leq 1$ h. Then, pIWP decreasing phase follows the same time evolution as Prec reaching minima ($Prec_{min}$ and $pIWP_{min}$) around

07:00-09:00 LT. Thus, ice in the troposphere follows the same diurnal cycle as the Prec within a time resolution of one hour.

In the TL, IWC is still proportional to Prec but IWC starts to increase later than $\left(\delta t_{on}^{TL} \sim 1h\right)$ the pIWP in the UT (Fig. 11a). After $t_{end}$, IWC decreases more rapidly than Prec and varies between +2 and -7 % compared to the mean between 23:00 and 09:00 LT. Thus, as soon as the deep convection reaches one of these two levels, ice is brought by the updraft during the growing and mature stage of the convective activity. Then, ice decreases with the downdraft of the convective dissipating

stage. However, the very deep convective activity reaching the TL starts later than the deep convection reaching the UT ($\delta t \approx 1$ h) and decreases quickly before to reach a background IWC minimum between 23:00 LT and 09:00 LT.

Over ocean during the growing phase (Fig. 11b), pIWP also increases proportionally to Prec. Assuming that $IWC^{146}$ evolves like pIWC, this implies that $t_{on}^{UT} \approx t_{on} + \delta t_{on}^{UT}$ $\left(\delta t_{on}^{UT} \geq 1h\right)$ at ~21:00-22:00 LT, and $t_{end}^{UT} \approx t_{end} + \delta t_{end}^{UT}$ at ~ 04:00-05:00 LT with $\delta t_{end}^{UT} > 1$ h. Thus, as $\delta t_{on}^{UT} \approx \delta t_{end}^{UT}$, the diurnal cycle of pIWP over ocean is synchronized with the diurnal cycle of Prec but

delayed. In the TL and during the growing phase, IWC starts to increase later than pIWP (where $\delta t_{on}^{TL} \approx 15$ h, Fig. 11b). This result is consistent with the diurnal cycle of OPF presented in Figure 9, where OPF starts to increase with a delay $\delta t_{on}^{TL} \approx 9$ h with respect to the increase of Prec. According to Liu and Zipser (2005), only less than 5% of OPF reach the TL, so other





processes in addition to deep convection govern the ice diurnal cycle in the TL over ocean impacting on the delay between

the onset of ice injection $t_{on}^{TL}$ and the onset of the convection $t_{on}$. After $t_{on}^{TL}$, IWC takes around 13 h to reach a maximum at

$t_{end}^{TL} \sim$ 19:00-20:00 LT.

Table 2 presents the magnitude of diurnal variation of ice, that is to say, the amount of ice injected into the UT and the TL

estimated using the model presented in section 4.2 from the MLS and TRMM measurements ($\Delta IWC^B$) and the magnitude of

diurnal variation of ice measured by SMILES ($\Delta pIWP$ and $\Delta IWC_{SMILES}$) over the six study zones and during DJF 2009-2010.

$\Delta IWC^{UT}$ and $\Delta IWC^{TL}$ are close to $\Delta pIWP_{SMILES}$ and $\Delta IWC_{SMILES}$, respectively (with differences between 0.02 to 0.28 mg m$^{-3}$).

Thus, during the ENSO period of DJF 2009-2010, the amount of ice injected in the UT ($\Delta IWC^{UT}$) is about 2.06-2.34 mg m$^{-3}$

over land and 1.20-1.22 mg m$^{-3}$ over ocean and in the TL ($\Delta IWC^{TL}$) is about 0.26-0.31 mg m$^{-3}$ over land and 0.13-0.22 mg

m$^{-3}$ over ocean. NOAA Interpolated Outgoing Longwave Radiation (OLR) during DJF 2009-2010 is consistent with the fact

that the convective activity grows higher over land than over ocean (not shown).

In summary, our method based on the correlation between Prec from TRMM and IWC from MLS during the growing stage

of the convection (presented in section 4.1) allows to estimate the amount of ice injected into the UT and the TL ($\Delta IWC^B$)

and has been validated with the SMILES measurements over land and ocean. These results confirm that the diurnal cycle of

Prec can be used as proxy of deep convection reaching the UT and the TL during the growing stage of the deep convection.

The analyses of SMILES data also suggest a good correlation of IWC$^{UT}$ and Prec during the dissipating stage of the

convection in the UT over land.

**4.3 Amount of ice injected in the austral convective UT**

Using the method presented in the previous section, 13 years of MLS and TRMM observations are analysed. Table 3

presents the time of the beginning of the IWC injection in the UT $\left(t_{on}^{UT}\right)$, the duration of the IWC injection $\left(\Delta t^{UT} = t_{end} - t_{on}^{UT}\right)$,

the minimum value of IWC before the beginning of the IWC injection $\left(IWC_{min}^{UT}\right)$ and the amount of ice injected in the UT

(equivalent to the amplitude of the IWC diurnal cycle, $\Delta IWC^{UT} = IWC_{max}^{UT} - IWC_{min}^{UT}$ ).

Over land, $t_{on}^{UT}$ is found during the morning at 08:00-09:00 LT whatever the land zone. However, $\Delta t^{UT}$ is longer over

MariCont-L than SouthAm and SouthAfr ($\Delta t^{UT}$ = 8, 7 and 7 h, respectively). Furthermore, $IWC_{min}^{UT}$ is lower over the

MariCont-L than over SouthAm and SouthAfr ( $IWC_{min}^{UT}$ = 1.04, 1.65 and 1.33 mg m$^{-3}$, respectively), but $\Delta IWC^{UT}$ is greater

over the MariCont-L than over SouthAm and SouthAfr ( $\Delta IWC^{UT}$ = 3.34, 1.99 and 2.86 mg m$^{-3}$, respectively).

Over ocean, $t_{on}^{UT}$ is found during the evening at 17:00-18:00 LT. $\Delta t^{UT}$ is much longer over MariCont-O than over IndOc and

PacOc ($\Delta t^{UT}$ = 11, 9, 9 h, respectively). However $IWC_{min}^{UT}$ ,is greater over MariCont-O than over IndOc and PacOc ($IWC_{min}^{UT}$ =

1.62, 1.01 and 1.46 mg m$^{-3}$ respectively) and $\Delta IWC^{UT}$ over MariCont-O is also greater than $\Delta IWC^{UT}$ over PacOc and IO

$\left( \Delta IWC^{UT} = 0.91, 0.46 \text{ and } 0.43 \text{ mg m}^{-3}, \text{ respectively} \right).$



Thus, the convective growing stage is quicker over land (7 h) than over ocean (10 h). These results are consistent with previous study from Takahashi and Luo (2014) showing that the time life of the mature stage of the deep convection is

between 6 and 12 h after the onset of the deep convection considering North and South hemispheres over the tropics. Furthermore, while the estimated means of $IWC_{min}^{UT}$ over land and ocean are very close (1.34 and 1.36 mg m$^{-3}$, respectively), the means of the land and oceans $\Delta IWC^{UT}$ how great differences (2.73 and 0.60 mg m$^{-3}$, respectively). Thus, the ice background is similar over land and ocean in the UT but there is 4 times more ice injected in the UT over land than over ocean. These results also show that the MariCont region is a region with greater amount of ice injection in the UT compared

to other tropical deep convective areas.

### 4.4 Amount of ice injected in the austral convective TL

From the method presented previously and the results from the SMILES instruments (Fig. 11), Table 4 presents $t_{on}^{TL}$, $\Delta t^{TL}$, $IWC_{min}^{TL}$ and $\Delta IWC^{TL}$ with respect to time for ice to be injected into the TL after the onset of deep convection, $\delta t^{TL}$ over the different study zones. We propose to use $\delta t^{TL} = 0$ to 3 h in order to keep $\delta t^{TL}$ always shorter than $t_{on}-t_x$, where $x$ is selected

during the growing phase ($x$=D over land and $x$=N over ocean) in order to perform a sensitivity study of our results.

Over land, for all $t_{on}^{TL}$, the period of injection ($\Delta t^{TL}$) is longer over MariCont-L than over SouthAm and SouthAfr ($\Delta t^{TL}$ = 8, 7 and 7 h, respectively). Depending on the $\delta t^{TL}$ (between 0 and 3 h), $\Delta t^{TL}$ can be between 7-8 and 4-5 h. $\Delta IWC^{TL}$ is greater over MariCont-L than over SouthAm and SouthAfr for all the $\delta t^{TL}$ ($\Delta IWC^{TL}$ ~ 0.56-0.42, 0.26-0.13 and 0.40-0.24 mg m$^{-3}$, respectively). As observed in the short time period study with SMILES, the injection of ice over MariCont-L is the most

intense over tropics. Furthermore, the smaller $\delta t^{TL}$, the longer $\Delta t^{TL}$, the greater $\Delta IWC^{TL}$.

Over ocean, as observed with the SMILES measurements, $t_{on}^{TL}$ cannot be estimated with our method. However, our method is able to estimate $\Delta t^{TL}$, $IWC_{min}^{TL}$ and $\Delta IWC^{TL}$ as a function of $\delta t^{TL}$ (as demonstrated in section 4.1). Over MariCont-O, ice injection in the TL is longer than over IndOc and PacOc ($\Delta t^{TL}$ = 8-11, 6-9 and 6-9 h, respectively). The amount of ice injected over MariCont-O is greater than over IndOc and PacOc ($\Delta IWC^{TL}$ = 0.16-0.19, 0.09-0.10, 0.08-0.09 mg m$^{-3}$,

respectively). Furthermore, the ice background in the TL is also greater over MariCont-O than over IndOc and PacOc ($IWC_{min}^{TL}$ ~ 0.34-0.37, 0.24-0.26, 0.29-0.30 mg m$^{-3}$, respectively). Thus, while the diurnal cycle of Prec over MariCont-O is the weakest, the diurnal cycle of ice over MariCont-O would have the highest values.

As expected from the SMILES measurements, the MariCont region presents the largest ice injection in the TL. While the IWC background over MariCont-O is the largest, the injection of IWC over MariCont-L is the most important. Furthermore,

all these results are consistent with the OLR showing strongest values over region having highest OLR signal (not shown).

### 5. Discussion on the processes driving the decreasing phase of the IWC diurnal cycle



The previous section has shown that the growing phase of the diurnal cycle of deep convection in the tropics impacts on the growing phase of diurnal cycle of ice in the UT and the TL. In this section, the convective processes impacting the

decreasing phase of the ice diurnal cycle in the UT and the TL are discussed. The decreasing phase of the diurnal cycle of ice in the UT and the TL represents the diurnal loss of ice. This loss can be caused by several processes including sedimentation, convective processes during the dissipating stage, sublimation into water vapour or horizontal advection and mixing. Quantifying the processes impacting the diurnal cycle of ice can help to quantify the amount of water staying into the UT and TL, the part falling down to the surface and the part advected up to the LS at the temporal diurnal scale.

In this section, the comparison between the value of IWC measured by MLS during the decreasing phase ($IWC_x$) at (x = Day or Night) and IWC estimated at the same $t_x$ ($IWC(t_x)$) is discussed. If $IWC_x \sim IWC(t_x)$, our model based on Prec evolution is a good predictive tools to consider that convective processes (as represented by Prec) control the decreasing phase of the ice diurnal cycle. Otherwise, non-convective processes has also need to be investigated to explain the ice diurnal cycle. Table 5 presents the difference $d$ ($d = IWC(t_x) – IWC_x$) in the UT and the TL.

Over land in the UT, $d$ is positive and less than 23 % over SouthAm and SouthAfr and close to zero over the MariCont-L. The fact that the amount of Prec over SouthAm and SouthAfr is greater than required to explain IWC suggests that a significant fraction of the convection is shallow and does not reach the UT. In contrast, over the MariCont-L, the diurnal cycle of Prec is very similar to the diurnal cycle of IWC suggesting that the convective activity over this region brings a large amount of ice all over the 24 hours. In the TL, $d$ is found between -14 and -20 % implying that the ice decreases more

slowly than Prec.

Over ocean in the UT, $d$ is negative down to -26 % showing that the ice stay longer in the UT compared to the Prec decreasing phase. In the TL, $d$ varies between +9 and -7 showing low differences $d$ at this level.

In summary, the method presented in the present paper allows to estimate the diurnal cycle of ice in the UT and the TL to within 26 % over the 6 tropical zones and to within 14 % over the MariCont-L region.

## 6. Conclusion and perspectives

To quantify the amount of ice injected in tropical upper troposphere (UT) and the tropopause layer (TL) and the processes linked to the ice diurnal variation is important to better understand the amount of total water in these layers and the amount of water entering into the LS. The information given by MLS on the diurnal cycle of ice water content (IWC) in the UT and

the TL and the method comparing the two per day measurements (Liu and zipser, 2005 and Carminati et al., 2014) are too limited to estimate the ice variability in these layers. Thus, the present study starts to assume that the diurnal cycle in ice is related to the diurnal cycle in precipitation (for which there are TRMM data with much better time resolution) in austral convective tropics. The high relationship between precipitation (Prec) measured by TRMM and IWC measured by MLS in the UT and the TL is shown in the study. As MLS measures IWC only twice a day, at 01:30 (Night) and 13:30 LT (Day), we

have proposed a simple model based on the diurnal cycle of Prec to estimate the diurnal cycle of IWC in the UT and in the



TL. The method is validated using the short period of DJF 2009-2010 of ice measurement from the SMILES instrument in the troposphere and the TL.

Because of the strong space correlation between Prec and IWC, we show that Prec can be used as a proxy of the deep convection reaching the UT and the TL during the growing stage of the convection in order to bring ice in the UT and the

TL. The method proposed in the study allows to estimate 1) the diurnal cycle of ice into the UT and the TL and 2) the difference between the maximum and the minimum in the diurnal cycle of ice in these layers, namely the amount of ice injected by deep convection. Our method suggests that deep convection is the most important process in austral convective tropics driving the diurnal increase of ice in these layers. Other processes may play a minor role such as the decrease of the temperature in the TTL, increasing the saturation ratio and allowing the crystal nucleation and growth, or the bring of ice in

the UT and TL by horizontal advection.

The amount of ice injected in the UT (2.73 mg m$^{-3}$ over land and 0.60 mg m$^{-3}$ over ocean) is found to be much higher than the amount of ice injected in the TL (0.26-0.41 mg m$^{-3}$ over land and 0.11-0.13 mg m$^{-3}$ over ocean). Furthermore, the study highlights the importance to separate the land of the Maritime Continent (MariCont-L) to the ocean of the Maritime Continent (MariCont-O) to better understand the ice diurnal cycle in the UT and the TL over the complex and strong

convective region of the Maritime Continent. It has been shown that the injection of ice over the MariCont-L into the UT and the TL ($\Delta IWC^{UT}$ = 3.34 mg m$^{-3}$, $\Delta IWC^{TL}$ = 0.56-0.42 mg m$^{-3}$) is the greatest in the tropics.

The decreasing phase of the ice diurnal cycle is also evaluated with Prec and discussed. Prec is also considered to be a good proxy of the decreasing phase of the convection to within 26 %, especially over the MariCont-L region in the UT (to within 3 %). In summary, the ice diurnal cycle in the UT and the TL is mainly governed by vertical processes linked to the convective

activity that are much stronger than other processes such as the horizontal mixing, the sublimation, the sedimentation.

Although it is beyond the scope of the present paper, some potentially new results might be achievable using our method as e.g. the part of ice sublimating during deep convection, the impact of intra-seasonal variability such as ENSO and the Madden Jullian Oscillation (MJO) on the ice diurnal cycle. In fact, ENSO and MJO have a strong influence on the tropical deep convection and injection of moisture into the TTL (Gettelman et al., 2002; Wong and Dessler, 2007) and are associated

to the cold anomalies near the TTL, especially over the Maritime Continent region (Zhang et al., 2013). We are expecting a strong impact on the diurnal ice injection timing and duration in the TL. The processes impacting the ice injection into the UT and the TL over the Maritime Continent regions at local scale will be studied in a forthcoming paper.

**Acknowledgment**

Our study takes place within the Turbulence Effects on Active Species in Atmosphere (TEASAO – http://www.legos.obs-mip.fr/projets/axes-transverses-processus/teasao) project. We thank the National Center for Scientific Research (CNRS) and the Excellence Initiative (Idex) of Toulouse, France (TEASAO project, Peter Haynes Chair of Attractivy) to fund this study. We would like to thank the teams that have provided the MLS data (https://disc.gsfc.nasa.gov/datasets/ML2IWC_V004), the



TRMM data (https://pmm.nasa.gov/data-access/downloads/trmm) and the SMILES data
(https://mls.jpl.nasa.gov/data/smiles).

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





**Figure 1: (From top to bottom): (a) (Day+Night)/2 of precipitation measured by TRMM at 0.25°×0.25° resolution (mm/h), (b) (Day+Night)/2 of precipitation at 2°×2° resolution (mm/h), and (c) hour of diurnal maximum of precipitation at 0.25°×0.25° resolution (h in Local Solar Time, LST) in DJF over the period 2004-2017.**





660

670






**Figure 2: (From top to bottom): (a) daily average (Day+Night)/2 of Ice Water Content (IWC), (b) Water Vapour (WV), (c) IWC fraction (IWC/∑W), (d) Temperature and (e) Relative Humidity with respect to Ice (RHI) in the upper troposphere (146 hPa) measured by MLS over the tropics in DJF over the period 2004-2017.**





**Figure 3: (From top to bottom): (a) daily average (Day+Night)/2 of Ice Water Content (IWC), (b) Water Vapour (WV), (c) IWC fraction (IWC/∑W), (d) Temperature and (e) Relative Humidity with respect to Ice (RHI) in the tropopause layer (100 hPa) measured by MLS over the tropics in DJF over the period 2004-2017.**







**Figure 4: Representation of the study zones: PacOc, Pacific Ocean; SouthAm, South America; SouthAfr, South Africa; IndOc, Indian Ocean; MariCont-L, land of the Maritime Continent; MariCont-O, ocean of the Maritime Continent. Each zone is defined at a horizontal resolution of 2°×2°.**






**Figure 5: Vertical profiles of (a) temperature, (b) Water Vapour (WV), (c) Ice Water Content (IWC) and (d) Relative Humidity with respect to Ice (RHI) from MLS data averaged from 2004 to 2017 in DJF over the 6 study zones: IndOc (light blue), PacOc (dark blue), SouthAfr (green), SouthAm (yellow), Tropics (black), MariCont-L (red solid line) and MariCont-O (red dashed line).**






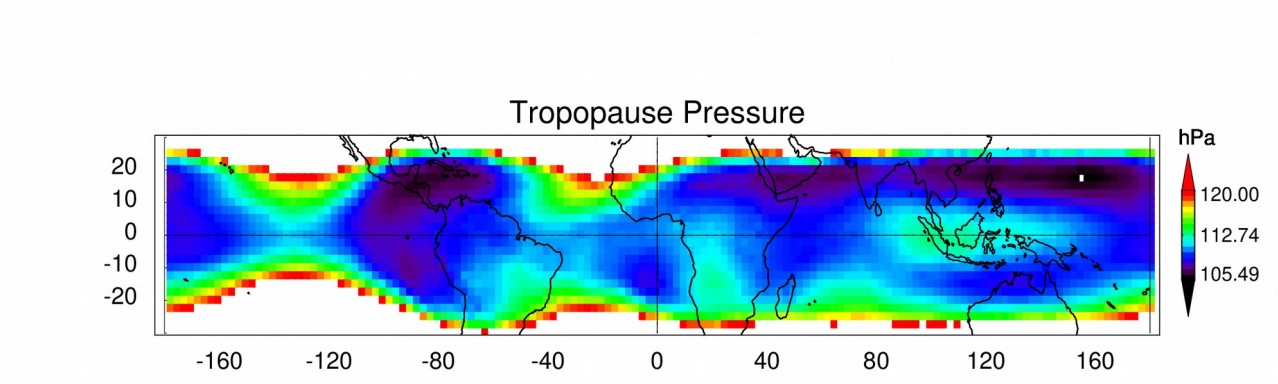

**Figure 6: Pressure of the tropopause (hPa) over the tropics during DJF from 2004 to 2017, defined from the NCEP datasets.**









**Figure 7: Day−Night of (a) precipitation (Prec) from TRMM, (b) Ice Water Content (IWC) from MLS at 146 hPa and (c) Water Vapour (WV) from MLS at 146 hPa over the tropics and the period 2004-2017 in DJF.**










**Figure 8:** **Diurnal cycle of precipitation (Prec) over (a) the land study zones and (b) the ocean study zones during the period DJF 2004-2017.**





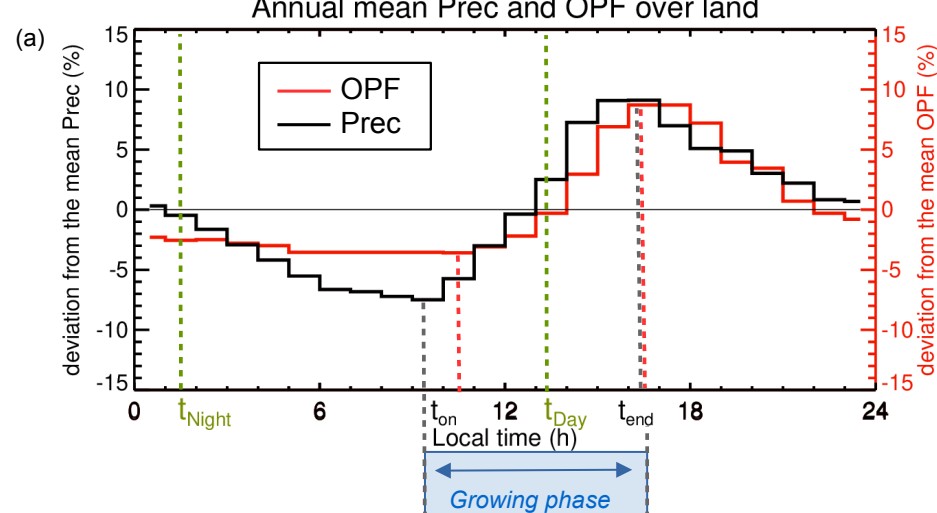

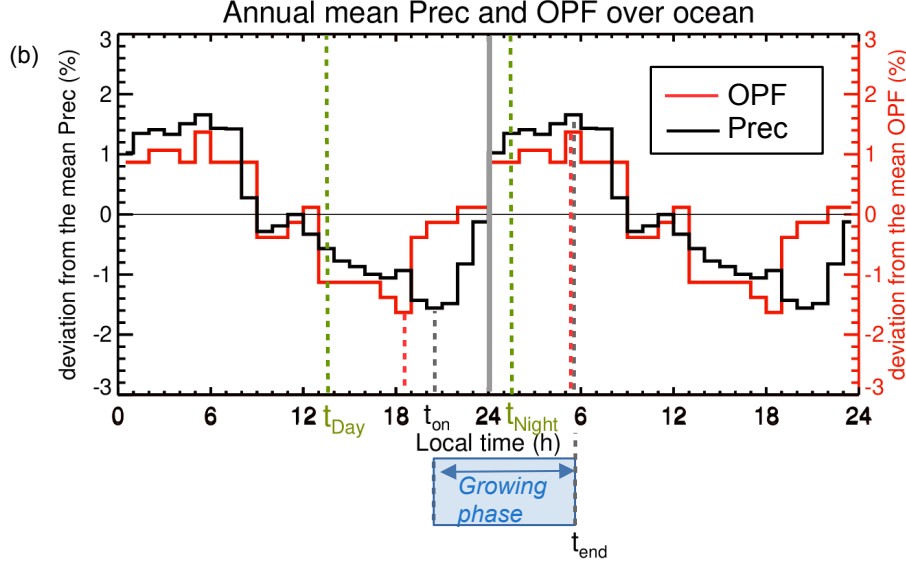

**Figure 9: (Top) Deviation from the mean of the annual mean diurnal cycle of TRMM precipitation (Prec, in black solid line) over land, from 2004 to 2017 and diurnal cycle of overshooting precipitation features (OPFs) at the tropopause (red solid line) over land, from 1998 to 2000 and from 2001 to 2003 adapted from Liu and Zipser (2005) over the tropics (20°N-20°S). (Bottom) Same as top but over the ocean. Minimum and maximum of Prec and OPF are drawn with vertical dotted lines (in black and red, respectively). $t_{on}$ is the time of the onset of the Prec, $t_{end}$ is the time of the diurnal maximum of Prec, $t_{Day}$ and $t_{Night}$ are the times of the two MLS measurements at Day and Night, respectively.**





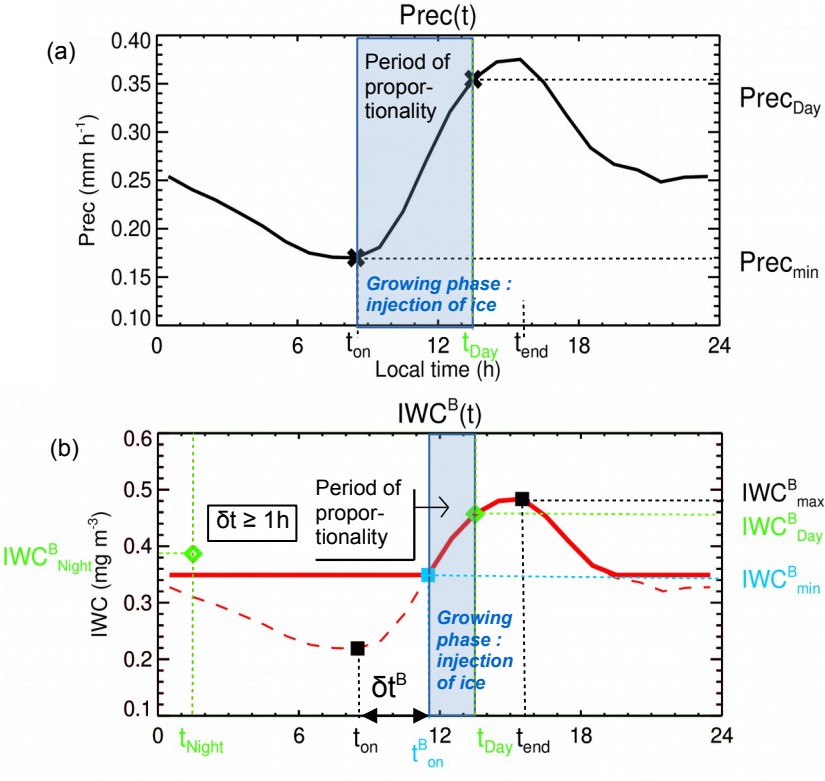

**Figure 10: Methodology to estimate the diurnal cycle of ice water content (IWC) over land: (a) diurnal cycle of precipitation, with ✖ representing the minimum of precipitation $\left(Prec_{min}\right)$ with its associated time $\left(t_{on}\right)$ and Prec during the day $\left(Prec_{Day}\right)$ at $t_{Day}$ = 13:30 LT. (b) Diurnal cycle of $IWC^{B}(t)$ in red solid lines, estimated from the diurnal cycle of Prec and from the two MLS measurements of ice ( ◆ , $IWC_{x}^{B}$), with the timing of the onset of the ice injection at $t_{on}^{B}=t_{on}+\delta t^{B}$, and ■ representing the IWC minimum $\left(IWC_{min}^{B}\right)$ when $\delta t^{B}$ < 1 h and the IWC maximum $\left(IWC_{max}^{B}\right)$ and ■ , the $IWC_{min}^{B}$ when $\delta t^{B}$ > 1h.**







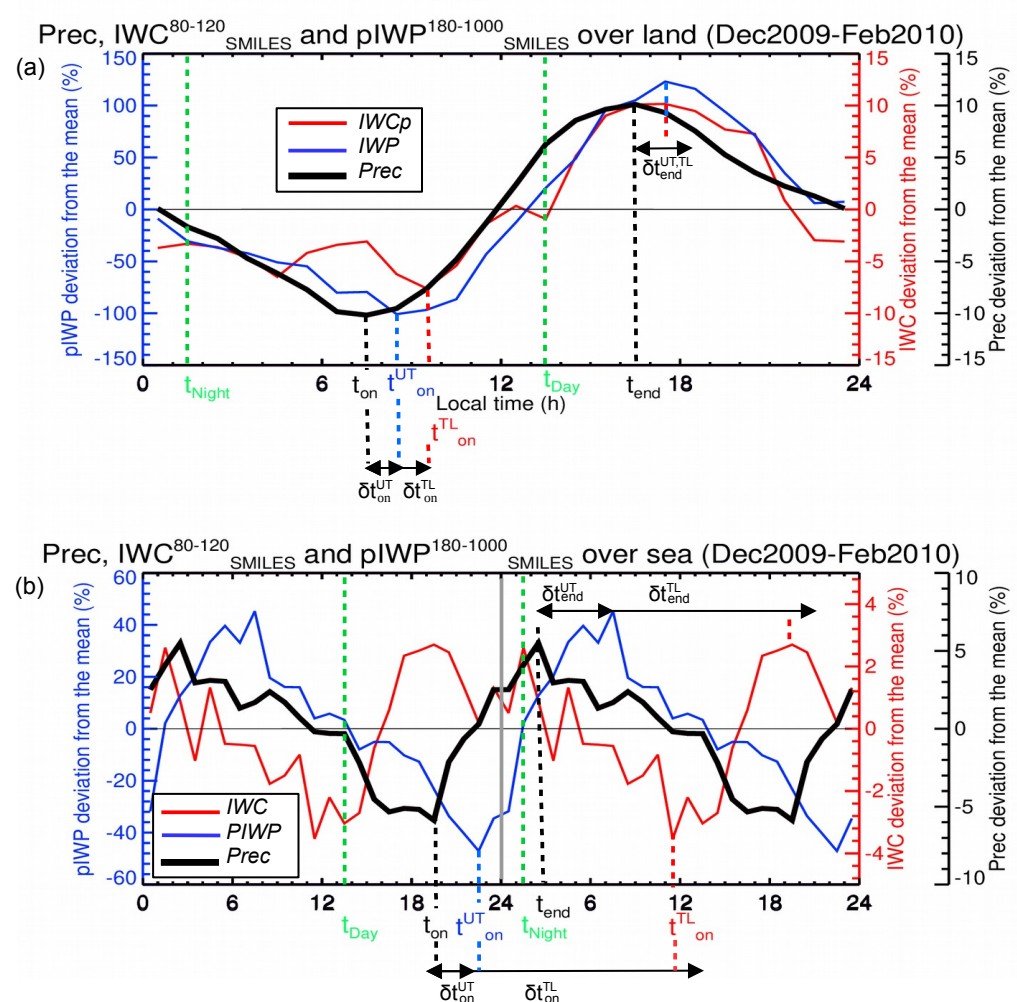

**Figure 11: Deviation from the mean (%) of the diurnal cycle of partial Ice Water Path (pIWP, integrated between 1000 and 180 hPa), Ice Water Content (IWC, averaged between 80 and 120 hPa) measured by SMILES and precipitation (Prec) over (a) the tropical land (0°S-30°N) and (b) tropical ocean (0°N-30°S) from December 2009 to February 2010. $t_{on}$, $t_{on}^{UT}$ and $t_{on}^{TL}$ are the onset of**

**the Prec, pIWP and IWC increase, respectively. $t_{end}$ is the end of the Prec increase. $\delta t_{on}^{UT}$ and $\delta t_{on}^{TL}$ are the delay between $t_{on}$ and**

**$t_{on}^{UT}$ and $t_{on}^{TL}$, respectively and $\delta t_{end}^{UT}$ and $\delta t_{end}^{UT}$ are the delay between $t_{end}$ and the end of the pIWP and IWC increase, respectively.**

**All the date have been filtered with a 6-h running average.**





**Table 1: Pearson linear correlation coefficient $R$, regression coefficient $\alpha$ (mg m$^{-3}$ mm$^{-1}$ h) and the offset $\beta$ (mg m$^{-3}$) from the correlation between precipitation and Ice Water Content (IWC), and Water Vapour (WV) in the upper troposphere (UT, at 146 hPa), at Day, Night and Day−Night over the 7 tropical zones and the average over the 6 zones ($\mu_{6Zones}$).**

|  |  | Land | | | Ocean | | | | |
|---|---|---|---|---|---|---|---|---|---|
|  |  | SouthAm | SouthAfr | MariCont-L | IndOc | PacOc | MariCont-O | MariCont | $\mu_{6Zones}$ |
| **IWC** | $R\ (IWC^{UT}_{Day}, P_{Day})$ | 0.7 | 0.8 | 0.5 | 0.8 | 0.8 | 0.6 | 0.5 | 0.7 |
|  | $\alpha\ (IWC^{UT}_{Day}, P_{Day})$ | 8.1 | 15 | 9.3 | 4.8 | 5.9 | 7.5 | 8.9 | 8.4 |
|  | $\beta\ (IWC^{UT}_{Day}, P_{Day})$ | 0.9 | 0.0 | 1.1 | 0.3 | 0.4 | 0.7 | 1.0 | 0.6 |
|  | $R\ (IWC^{UT}_{Night}, P_{Night})$ | 0.7 | 0.7 | 0.6 | 0.9 | 0.7 | 0.6 | 0.6 | 0.7 |
|  | $\alpha\ (IWC^{UT}_{Night}, P_{Night})$ | 4.8 | 6.1 | 5.6 | 4.7 | 6.8 | 7.7 | 5.4 | 5.9 |
|  | $\beta\ (IWC^{UT}_{Night}, P_{Night})$ | 0.6 | 0.6 | 1.4 | 0.3 | 0.2 | 0.6 | 1.6 | 0.6 |
|  | $R\ (IWC^{UT}_{Day-Night}, P_{Day-Night})$ | 0.6 | 0.7 | 0.7 | 0.1 | 0.2 | 0.5 | 0.5 | 0.5 |
| **WV** | $R\ (WV^{UT}_{Day}, P_{Day})$ | 0.4 | 0.3 | 0.2 | 0.0 | 0.3 | 0.1 | 0.2 | 0.2 |
|  | $\alpha\ (WV^{UT}_{Day}, P_{Day})$ | 0.0 | 0.0 | 0.0 | 0.0 | 0.0 | 0.0 | 0.0 | 0.0 |
|  | $R\ (WV^{UT}_{Night}, P_{Night})$ | 0.5 | 0.5 | 0.1 | 0.0 | 0.5 | 0.0 | 0.0 | 0.3 |
|  | $\alpha\ (WV^{UT}_{Night}, P_{Night})$ | 0.0 | 0.0 | 0.0 | 0.0 | 0.0 | 0.0 | 0.0 | 0.0 |
|  | $\mathbf{R\ (WV^{UT}_{Day-Night}, P_{Day-Night})}$ | -0.0 | -0.1 | 0.2 | 0.2 | 0.0 | 0.1 | 0.2 | 0.1 |





**Table 2: Differences between the amount of Ice Water Content (IWC) estimated from TRMM and MLS injected in the upper troposphere (UT) and the tropopause layer (TL) ($\Delta IWC^{UT}$ and $\Delta IWC^{TL}$, respectively), and partial Ice Water Path (pIWP) measured by SMILES in the UT ($\Delta pIWP_{SMILES}$) and IWC measured by SMILES in the TL ($\Delta IWC_{SMILES}$), during the period DJF 2009-2010, and their differences ($\Delta IWC^{UT}$-$\Delta pIWP_{SMILES}$ and $\Delta IWC^{TL}$-$\Delta IWC_{SMILES}$, in the UT and the TL, respectively). The average over the land and ocean zones ($\mu$(Lands ZonesTropics), $\mu$(Oceans ZonesTropics), respectively) are also presented.**

| | | UT | | | TL | | |
|---|---|---|---|---|---|---|---|
| | | $\Delta IWC^{UT}$ (mg m$^{-3}$) | $\Delta pIWP_{SMILES}$ (mg m$^{-3}$) | $\Delta IWC^{UT}$- $\Delta pIWP_{SMILES}$ (mg m$^{-3}$) | $\Delta IWC^{TL}$ (mg m$^{-3}$) | $\Delta IWC_{SMILES}$ (mg m$^{-3}$) | $\Delta IWC^{TL}$- $\Delta IWC_{SMILES}$ (mg m$^{-3}$) |
| L A N D | SouthAm | 2.13 | 2.85 | -0.72 | 0.34 | 0.29 | +0.05 |
| | SouthAfr | 1.53 | 1.85 | -0.32 | 0.23 | 0.27 | -0.04 |
| | MariCont-L | 2.52 | 2.33 | +0.19 | 0.35 | 0.23 | +0.12 |
| O C E A N | IndOc | 1.13 | 1.01 | +0.12 | 0.34 | 0.09 | +0.15 |
| | PacOc | 1.05 | 1.56 | -0.51 | 0.21 | 0.24 | -0.03 |
| | MariCont-O | 0.49 | 1.05 | -0.56 | 0.10 | 0.08 | +0.02 |
| | $\mu$(Lands ZonesTropics) | 2.06 | 2.34 | - 0.28 | 0.31 | 0.26 | +0.05 |
| | $\mu$(Oceans ZonesTropics) | 1.22 | 1.20 | +0.02 | 0.22 | 0.13 | +0.09 |




**Table 3: Time of the onset of the ice injection in the tropical upper troposphere (UT), $(t_{on}^{UT})$, duration of the injection of ice in the UT $(\Delta t^{UT})$, minimum amount of ice in the UT $(IWC_{min}^{UT})$, and amount of ice injected into the UT $(\Delta IWC^{UT})$ as a function of the 6 study zones and averaged over the land and ocean zones ($\mu$(Lands ZonesTropics), $\mu$(Oceans ZonesTropics), respectively) during DJF from 2004 to 2017. The bolded values highlight the most important injection of ice and the associated regions.**


| Region | $t^{UT}_{min}$ (LT) | $\Delta t^{UT}$ (h) | $IWC^{UT}_{min}$ (mg m$^{-3}$) | $\Delta IWC^{UT}$ (mg m$^{-3}$) |
|---|---|---|---|---|
| SouthAm | 08:00-09:00 | 7 | 1.65 | 1.99 |
| SouthAfr | 08:00-09:00 | 7 | 1.33 | 2.86 |
| **MariCont-L** | 08:00-09:00 | 8 | 1.04 | **3.34** |
| IndOc | 17:00-18:00 | 9 | 1.01 | 0.43 |
| PacOc | 17:00-18:00 | 9 | 1.46 | 0.46 |
| **MariCont-O** | 17:00-18:00 | 11 | 1.62 | **0.91** |
| *$\mu$(Lands ZonesTropics)* | 08:00-09:00 | 7 | 1.34 | 2.73 |
| *$\mu$(Oceans ZonesTropics)* | 17:00-18:00 | 10 | 1.36 | 0.60 |








**Table 4: As a function of the delay ($\delta t = 0, 1, 2,$ or 3 h) between the beginning of the Prec onset and the IWC onset in the tropical tropopause layer (TL): time of the onset of the ice injection in the TL $\left(t_{on}^{TL}\right)$, duration of the injection of ice in the TL $\left(\Delta t^{TL}\right)$, minimum amount of ice in the TL $\left(IWC_{min}^{TL}\right)$ and amount of ice injected into the TL $\left(\Delta IWC^{TL}\right)$ as a function of the 6 study zones and averaged over the land and ocean zones ($\mu$(Lands ZonesTropics), $\mu$(Oceans ZonesTropics), respectively) during DJF from 2004**

**to 2017. The bolded values highlight the most important $IWC_{min}^{TL}$ and $\Delta IWC^{TL}$ and the associated regions.**

| Zones | $\delta t = 0$ h | | | $\delta t = 1$ h | | | $\delta t = 2$ h | | | $\delta t = 3$ h | | |
|---|---|---|---|---|---|---|---|---|---|---|---|---|
| | $\Delta t^{TL}$ | $IWC_{min}^{TL}$ | $\Delta IWC^{TL}$ | $\Delta t^{TL}$ | $IWC_{min}^{TL}$ | $\Delta IWC^{TL}$ | $\Delta t^{TL}$ | $IWC_{min}^{TL}$ | $\Delta IWC^{TL}$ | $\Delta t^{TL}$ | $IWC_{min}^{TL}$ | $\Delta IWC^{TL}$ |
| | (h) | (mg m$^{-3}$) | (mg m$^{-3}$) | (h) | (mg m$^{-3}$) | (mg m$^{-3}$) | (h) | (mg m$^{-3}$) | (mg m$^{-3}$) | (h) | (mg m$^{-3}$) | (mg m$^{-3}$) |
| SouthAm | 7 | 0.22 | 0.26 | 6 | 0.23 | 0.25 | 5 | 0.28 | 0.20 | 4 | 0.35 | 0.13 |
| SouthAfr | 7 | 0.19 | 0.40 | 6 | 0.21 | 0.38 | 5 | 0.26 | 0.33 | 4 | 0.35 | 0.24 |
| **MariCont-L** | 8 | 0.17 | **0.56** | 7 | 0.19 | **0.55** | 6 | 0.23 | **0.50** | 5 | 0.32 | **0.42** |
| IndOc | 9 | 0.24 | 0.10 | 8 | 0.24 | 0.10 | 7 | 0.25 | 0.09 | 6 | 0.26 | 0.09 |
| PacOc | 9 | 0.29 | 0.09 | 8 | 0.29 | 0.09 | 7 | 0.29 | 0.09 | 6 | 0.30 | 0.08 |
| **MariCont-O** | 11 | **0.34** | **0.19** | 10 | **0.35** | **0.18** | 9 | **0.36** | **0.17** | 8 | **0.37** | **0.16** |
| *μ(Lands ZonesTropics)* | 7 | 0.19 | 0.41 | 6 | 0.21 | 0.39 | 5 | 0.26 | 0.34 | 4 | 0.34 | 0.26 |
| *μ(Oceans ZonesTropics)* | 10 | 0.29 | 0.13 | 9 | 0.29 | 0.12 | 8 | 0.30 | 0.12 | 7 | 0.31 | 0.11 |







**Table 5: Difference *d* between Ice Water Content measured by MLS during the decreasing phase (*IWC$_x$*) and Ice Water Content estimated at the same hour (*IWC(x)*) in the upper troposphere (UT) and the tropopause layer (TL), (where *x* = Day or Night as a function of the timing of the decreasing phase of the ice diurnal cycle).**

| | Zones | *d* in the UT (%) | *d* in the TL (%) |
|---|---|---|---|
| **L A N D** | SouthAm | + 23 | - 20 |
| | SouthAfr | + 20 | - 18 |
| | MariCont-L | - 3 | - 14 |
| **O C E A N** | IndOc | - 21 | - 7 |
| | PacOc | - 10 | + 9 |
| | MariCont-O | - 26 | - 8 |







**Appendix 1: List of Abbreviations and Acronyms and Their Meanings**

| Acronyms | Meanings |
|---|---|
| CPT | Cold Point Tropopause |
| DJF | December, January, February |
| IndOc | Indian Ocean |
| IWC | Ice Water Content |
| LS | Lower Stratosphere |
| LT | Local Time |
| MariCont | Maritime Continent |
| MariCont-L | Maritime Continent land |
| MariCont-O | Maritime Continent ocean |
| MLS | Microwave Limb Sounder |
| Prec | Precipitation |
| pIWP | partial Ice Water Path |
| PacOc | Pacific Ocean |
| RHI | Relative Humidity with respect to Ice |
| SouthAm | South America |
| SouthAfr | South Africa |
| TEMP | Temperature |
| TL | Tropopause Layer |
| TRMM | Tropical Rain Measuring Mission |
| TST | Tropophere to Stratosphere Transport |
| TTL | Tropical Tropopause Layer |
| UT | Upper Troposphere |
| WV | Water Vapour |
