# Peer review of "Ice injected up to the Tropopause by Deep Convection: 1) in the Austral Convective Tropics"

_Atmospheric Chemistry and Physics, 2018_

## Referee Comment (RC1) · Anonymous Referee #2 · 24 Nov 2018

The study by Iris-Amata Dion and coauthors addresses a topical issue of water transport in the tropical tropopause layer during Austral convective season. The analysis makes use of satellite observations of ice water content (IWC) by MLS and precipitation (Prec) by TRMM in consideration of twice daily sampling of the former and a full diurnal coverage of the latter. Having established a reasonably high spatial correlation between IWC and Prec, the authors propose an original method for reconstructing the diurnal cycle of IWC from MLS observations, enabling a quantification of the diurnal amount of ice convectively injected into the TTL. The method is validated using partial path IWC measurements by SMILES instrument during an individual convective season. The result are reported separately at two levels representing upper troposphere (146 hPa) and tropopause level (100 hPa) as well as for different land and oceanic

zones in the southern tropics considered as most convective regions. The results of analysis indicate a much stronger convective transport of ice above continental regions compared to oceanic ones. The largest amount of injected ice is found for the land regions of Maritime continent. A major strength of this study is the synergistic approach making use of different satellite observation techniques, which enables acquiring add-on value information on the IWC diurnal cycle from the twice-daily MLS observations. The manuscript is logically structured, the description of data and methods is comprehensive and the graphical material is good quality. The conclusions are in line with what is established by the analysis, and overall the study represents a valuable contribution to the topic of convective transport of water. There are however certain shortcomings which require revision or clarification. The detailed remarks are listed below.

General remarks.

* The introduction places the study into the context of stratospheric water vapour and related problematics. With that, in my opinion, the obtained results are of limited relevance for the control of stratospheric water budget. The stratospheric entry of water is mostly driven by minimum temperature at the Cold Point Tropopause and, to a much smaller extent, by injection of ice into subsaturated environment above this level. The CPT level corresponds to 82 hPa level of MLS (as can be clearly seen in Fig. 5a), whereas the analysis is performed for 146 and 100 hPa levels. The ice water detrained below CPT may have very little or no impact on stratospheric water: even if the injected ice crystals sublimate before settling down, the amount of water vapour ascending into the stratosphere would be limited by the colder temperatures at CPT level. Moreover, large-scale convection may lead to additional cooling of CPT at diurnal and intra-diurnal time scales thereby further limiting the cross-tropopause transport of water. I believe the above considerations should be discussed in the context of the role of deep convection of stratospheric water. It would also be useful to compare the results regarding IWC obtained for 100 hPa with those for 82 hPa level.

* On the base of comparison between TRMM Prec and TRMM OPF diurnal cycles,

Prec is shown to be a good proxy of deep convection during its growing phase. Could you clarify why the OPF data as such could not be used for the IWC analysis?

Specific remarks.

l. 18-19. The purpose of the method is missing in this sentence, i.e. "We propose a method for... using..."

l. 165. The title of Sect. 3.1 should be more specific.

l. 180. The statement regarding a ubiquitous subsaturation in the Austral convective regions is surprising. Consider that the RH product of MLS may have a substantial dry bias at 82 hPa since it is based on MLS temperature profiles. The latter, in turn, do not resolve the sharp temperature minimum at CPT, which leads to underestimation of RHI at this level. While the geographical distribution of MLS RH could be fairly accurate, the quantitative statements based on these data should be avoided. A correct way to infer RHI values from MLS measurements would be to compute them from WV profiles of MLS and temperature profiles from a reanalysis data set with sufficient vertical resolution in the TTL, e.g. MERRA-2.

l.203. Since Fig. 5a shows the profiles at the native MLS pressure levels, a correct inference on the CPT pressure would be 83 hPa $\pm$ half width of MLS weighting function.

l.204-207 and Fig. 6. What is the relevance of this information in the context of Sect. 3.2 entitled "Water budget in the UT and LS"? Please clarify.

l.282-283. The section title refers to UT level, whereas the first line reads "...convection reaching the tropopause..."

Sect. 4.2. For consistency, the description of SMILES instrument should be introduced in Sect. 2. Please clarify how the full diurnal coverage is ensured with ISS platform.

Figure 1. Color scales of 1a and 1b should be the same. Figure 1c is a duplicate of 1a.

Technical corrections.

l.175. Maximal => maximum

l.213. Convectively-lifted?

l.221. Spatial correlation

l.223. Consider rephrasing the sentence ending with "...WV and IWC in the UT and Prec is analyzed".

l.237-238. Broken sentence

l.376. try "before reaching"

l.417 sentence unclear

l.879. 30 N => 30S

l.882. sentence unclear

---

## Referee Comment (RC2) · Anonymous Referee #1 · 7 Dec 2018

Review of

**Ice injected up to the Tropopause by Deep Convection:**
**1) in the Austral  Convective Tropics**

by Dion et al.

**General**

The study presents a detailed investigation of the diurnal cycle of  water vapor and especially ice injected by deep convection into the UT and TL of six tropical  regions. The region with the highest amout of UT/TL ice -and thus probably a major  source region for stratospheric water- is identified. For that purpose, a clever data evaluation method was developed combining the satellite observations of  MLS (lower geographical and time resolution) and TRMM (higher geographical and time resolution). The method was validated (or better say evaluated, see specific comments) using measurements from the SMILES instrument. The topic of the study  is of relevance and suitable for publication in ACP. The manuscript is clearly structured and fluently written, the Figures are also clear and mostly self-explanatory.

What I missed in the paper is a more conclusive statement about the importance  of the findings and the impact for the research in this area – what new can we learn from this study ? Though the reader can get an idea about this, to my opininion it should be cleary pointed out in the paper at the relevant places (abstract, introduction, conclusions).

A number of specific comments are listed in the following in the order of appearance in the manuscript.

**Specific comments**

**1) line 9**   ' The impact of deep convection on the water budget  ...'

Comment: Isn't it that in this study the magnitude of water and ice in the UT and TL stemming from deep convection  is investigated – to estimate the impact on the water budget you would need to know the magnitude of the other sources of ice, yes ?

**2) line 46**    '.. the region called Maritime Continent (MariCont), the region between the Indian Ocean and the West Pacific …'

Comment: Figure 4 can be introduced already here – then the reader see bettter what region is meant.

**3) line 120**  2 Datasets       Comment: I recommend to rename the section to 'Instruments'

**4) line 127**  'The MLS IWC sensitivity thresholds do not allow to detect low ice content such as cirrus outflow associated with convective events.  MLS IWC sensitivity will mostly be able to detect ice from convective cores. Following Livesey et al. (2017), we will not consider IWC measurements less than 0.02 mg m−3.'

Comment:  All the thin cirrus are missed. How does that impact the results ?

**5) line Figure 1:** Panels a) and c) look identical – is that true or a mistake ?

**6) line 173**   'In DJF, local maxima of Prec,  IWC, WV, IWC fraction and RHI in the UT (Fig. 2) ...'

Comment:  Prec is shown in Fig. 1, please note.

**7) line 185 ff**  'According to the difference between the UT and the TL, WV decreases more with altitude over the MariCont region compared to all other tropical regions.'  ...

Comment:   Can you further explain why WV decreases more with altitude over the MariCont ? 'According to the difference between the UT and the TL' – what do you mean with that ?

… 'Consistently, TEMP is lower at 100 hPa (near the CPT) than at 146 hPa and its value is the lowest over the MariCont region.'

Comment:  Isn't it vice versa: consistently with the lower TEMP over the MariCont region the WV is lower ?

**8) line 188**     'While WV decreases by more than 8 ppmv in the TL over the MariCont region compared to the UT, the RHI in the TL reaches high values (RHI ~100 %) highlighting a saturated environment ...'

Comment:  This is not a surprise – the high RHI is consistent with the low TEMP. Please note.

**9) line 192**   'To investigate the vertical distribution and the diurnal cycles of water species in the TL, we have defined seven tropical convective zones shown in Figure 4'

Comment:  It would be better to introduce Fig. 4 earlier (see comment on line 46).

**10) line 211**  'RHI is lower in the UT than at the TL by ~10 % '

Comment:  This is not a new finding – please provide references.

**11) line 213**   'The convective lifted ice does not sublimate as rapidly as in the UT ...'

Comment: in case of saturation or supersaturation, the ice does not sublimate at all … as you state later in the paragraph, existing ice crystals even grow and sediment.  I recommend to rephrase the sentence to provide a clear picture of the processes.

By the way:   dehydratation  →  dehydration (line 215)

**12) line Figure 5**   The water vapor axis in panel b), I would recommend to use a logarithmic scale to make the differences in the TL better visible.

**13) line 237**  'Over land, all regions under consideration show a better efficiency to inject ice in the UT during Day than during N. is βx  the IWC background.'

Comment:  Something is wrong with the end of the sentence (marked in blue).

**14) line 259**  'Deep convection does not inject directly WV in the UT'.

Comment:  Is that true ? Look for example the recently appeared paper:

Convective hydration in the tropical tropopause layer during the StratoClim aircraft campaign: Pathway of an observed hydration patch

Keun-Ok Lee et al.

https://www.atmos-chem-phys-discuss.net/acp-2018-1114/

**15) line 327**  ' … (the diagram over ocean would look different since maxima appear during local night ton ).is defined as the time …

Comment:  Something is wrong with the end of the sentence (marked in blue).

**16) line Figure 10**  What is the dashed red line in panel b) ?   **332** ' (see the period of proportionality in Fig. 10).'  Is this the dashed red line ? Please indicate.

**17) line 333** 'This hypothesis considers deep convection represented by Prec as the main process bringing ice into the UT and the TL.'

Comment:  Could  you please  further justify the hypothesis that deep convection is the main process bringing ice into the UT and the TL – what is with the thin cirrus that are not included in your analysis? They can also bring (or form) ice in these regions.

**18) line 332, 334**  P(t) and Px :  P means Prec ?

**19) line Figure 11**  The colors in the panels are differently defined:

> (a) red – IWCp , blue – IWP,  black – Prec

> (b)  red – IWC , blue – PIWP, black – Prec

In the caption (and text)  you define pIWP, IWC, Prec.  Please clarify.

**20) line 356**  4.2 Validation of the method with SMILES measurements

Comment:

(a) The SMILES instrument is important for the evaluation of the method, but it is not introduced (for example in Section 2) and no references are given.  Please insert a brief description of the IWC detection method and the uncertainties of the measurements to  convicingly  demonstrate   that SMILES is suitable to  evaluate  the data analysis method applied here.

(b) Validation : I suggest to better use 'Evaluation' , since also the SMILES measurements are no absolute proof.

**21) line 393**  'NOAA Interpolated Outgoing Longwave Radiation (OLR) during DJF 2009-2010 is consistent with the fact that the convective activity grows higher over land than over ocean (not shown).'

Comment:  What do you mean with 'consistent' ? Even if  a figure is not presented, the ranges of OLR over land and over ocean should be specified.

**22) line 417**  how = show

**23) line 423**  'Table 4 presents $t_{on}^{TL}$, $\Delta t^{TL}$, $IWC_{min}^{TL}$ and $\Delta IWC^{TL}$ ...'

Comment:  $t_{on}^{TL}$ is not shown in Table 4.

**24) line 440**  '… all these results are consistent with the OLR showing strongest values over region having highest OLR signal (not shown).'        ???

**25) line 447** '… convective processes during the dissipating stage,...'

Comment:  As long as convection is active, how can the ice clouds are in a dissipatig stage ? Or do you mean '… decreasing convective processes …' ?  See also line **452**.

**26) line 483**  'Other processes may play a minor role such as the decrease of the temperature in the TTL, increasing the saturation ratio and allowing the crystal nucleation and growth, or the bring of ice in the UT and TL by horizontal advection.'

Comment: How do you know that these processes are minor? Do you know their order of  magnitude in comparison to deep convection  ?   From my feeling  your conclusion is right, but for an ensured statement I think an estimate of the other processes is needed.

**27) line 491**  '.. the injection of ice over the MariCont-L into the UT and the TL … is the greatest in the tropics.'

Comment:  So your finding is that deep convection is the major process feeding the  'stratospheric fountain' and not large-scale three-dimensional circulation, yes ? You mentioned in the abstract that the relative importance of the two processes continues to be debated – so I think you could better pronounce the importance of you finding.  This is true also for line **495**.
* * *
Two papers that might be of interest for you study:

Robrecht et al. (2018): Mechanism of ozone loss under enhanced water vapour conditions in the mid-latitude lower stratosphere in summer, ACPD.

Smith et al. (2017): A Case Study of Convectively Sourced Water Vapor Observed in the Overworld Stratosphere over the United States, JGR.

---

## Author Comment (AC1) · 31 Jan 2019

The responses was uploaded in the form of a supplement.

Please also note the supplement to this comment:
https://www.atmos-chem-phys-discuss.net/acp-2018-1006/acp-2018-1006-AC1-supplement.pdf
* * *

---

## Author Comment (AC2) · 31 Jan 2019

**Discussion paper**
15ʳᵈ January 2019

**Manuscript Title:** Ice injected up to the Tropopause by Deep Convection: 1) in the Austral Convective Tropics **by Dion et al.**

We would like to thank the two referees for the comments that were helpful in improving substantially the presentation and contents of the revised manuscript. We hope we have addressed appropriately all issues raised by the referees.

The referee's comments are repeated in the following paragraphs in black and our responses appear in red. Copy of sentences or paragraphs changed into the paper are repeated and appear in yellow. Sentences or paragraphs deleted into the paper appear crossed-out and in orange.  A copy of the entire manuscript with the whole corrections is presented at the end of this report.

**RESPONSES TO THE ANONYMOUS REFEREE #1**

**General**
The study presents a detailed investigation of the diurnal cycle of water vapor and especially ice injected by deep convection into the UT and TL of six tropical regions. The region with the highest amout of UT/TL ice and thus probably a major source region for stratospheric water- is identified. For that purpose, a clever data evaluation method was developed combining the satellite observations of MLS (lower geographical and time resolution) and TRMM (higher geographical and time resolution). The method was validated (or better say evaluated, see specific comments) using measurements from the SMILES instrument. The topic of the study is of relevance and suitable for publication in ACP. The manuscript is clearly structured and fluently written, the Figures are also clear and mostly self-explanatory.  What I missed in the paper is a more conclusive statement about the importance of the findings and the impact for the research in this area – what new can we learn from this study ? Though the reader can get an idea about this, to my opininion it should be cleary pointed out in the paper at the relevant places (abstract, introduction, conclusions).

We provide some suggestions on that separately below. We also modified the entire introduction (copied at the end of this report) and provide more justification for our work.

A number of specific comments are listed in the following in the order of appearance in the manuscript.

**Specific comments**
**1) line 9** ' The impact of **deep convection** on the water budget ...'
Comment: Isn't it that in this study the magnitude of water and ice in the UT and TL stemming from deep convection is investigated – to estimate the impact on the water budget you would need to know the magnitude of the other sources of ice, yes?

Yes, to be clearer, the sentence l. 9 has been changed to:

>
> The contribution of deep convection to the amount of water vapor and ice in the Tropical Tropopause Layer (TTL) from the tropical upper troposphere (UT, around 146 hPa) to the

Tropopause Level (TL, around 100 hPa) is investigated. Ice water content (IWC) and water vapour (WV) measured in the UT and the TL by the Microwave Limb Sounder (MLS, Version 4.2) are compared to the precipitation (Prec) measured by the Tropical Rainfall Measurement Mission (TRMM, Version 007).

**2) line 46 '..** the region called Maritime Continent (MariCont), the region between the Indian Ocean and the West Pacific ...'
Comment: Figure 4 can be introduced already here – then the reader see better what region is meant.

We surrounded by a red box the MariCont in the Fig.1 and we added this sentence in the caption:

Figure 1: (From top to bottom): (a) (Day+Night)/2 of precipitation measured by TRMM at 0.25°×0.25° resolution (mm/h), (b) (Day+Night)/2 of precipitation at 2°×2° resolution (mm/h), and (c) hour of diurnal maximum of precipitation at 0.25°×0.25° resolution (h in Local Solar Time, LST) in DJF over the period 2004-2017. The red box surrounds the MariCont region.

And added line 199:

... together with the Maritime Continent region (MariCont, the region made of lands and oceans, between the Indian Ocean and the West Pacific, presented by the red box in Fig. 1).

**3) line 120** 2 Datasets Comment: I recommend to rename the section to 'Instruments'
We renamed the section as you suggested.

**4) line 127** 'The MLS IWC sensitivity thresholds do not allow to detect low ice content such as cirrus outflow associated with convective events. MLS IWC sensitivity will mostly be able to detect ice from convective cores. Following Livesey et al. (2017), we will not consider IWC measurements less than 0.02 mg m$^{-3}$.'
Comment: All the thin cirrus are missed. How does that impact the results ?

This sentence was not clear enough and have been changed. The detection limit of MariCont ice concentration is 0.02 mg/m$^{-3}$ in the TL whatever the origin/source of ice: convective cores or cirrus associated with convective events. Thus, cirrus are not missed.

The new sentence l. 127 is now l.135:

The MLS IWC sensitivity thresholds are 0.1 mg m$^{-3}$ in the UT and 0.02 mg m$^{-3}$ in the TL (Livesey et al., 2017) and will imply a small under-estimation of the IWC measured values.

Furthermore, we changed according the detection limits values in the Tables 2 and 3.

**5) Figure 1:** Panels a) and c) look identical – is that true or a mistake ?
Yes it was a mistake. We have changed the panel c) by the correct one.

**6) line 173** 'In DJF, local maxima of Prec, IWC, WV, IWC fraction and RHI in the UT (Fig. 2) ...'
Comment: Prec is shown in Fig. 1, please note.
We modified the sentence l. 295 as:

In DJF, local maxima of Prec (Fig. 1), IWC, WV, IWC fraction and RHI in the UT (Fig. 2) are found over the main convective areas

**7) line 185** 'According to the difference between the UT and the TL, WV decreases more with altitude over the MariCont region compared to all other tropical regions.' …
Comment: Can you further explain why WV decreases more with altitude over the MariCont?

WV (Figures 2b and 3b) decreases more with altitude because of the temperatures (Figures 2d and 3d). Like temperature, the saturation WV concentration decreases with altitude, leading to the decrease of the observed WV.

We answer to these questions in changing the text l.188. See answer #8) line 309.

Furthermore, we have changed the scales of Figures 2 and 3 a, b and d in order to have the same scale sample in the UT and the TL (the new figures are copied at the end of this report).

'According to the difference between the UT and the TL' – what do you mean with that ?
... 'Consistently, TEMP is lower at 100 hPa (near the CPT) than at 146 hPa and its value is the lowest over the MariCont region.'
Comment: Isn't it vice versa: consistently with the lower TEMP over the MariCont region the WV is lower ?

We answer to these questions in changing the text l.188. See answer #8) line 309.

**8) line 188** 'While WV decreases by more than 8 ppmv in the TL over the MariCont region compared to the UT, the RHI in the TL reaches high values (RHI ~100 %) highlighting a saturated environment ...' Comment: This is not a surprise – the high RHI is consistent with the low TEMP. Please note.

We have changed by the following sentences starting line 309:

~~According to the difference between the UT and the TL, WV decreases more with altitude over the MariCont region compared to all other tropical regions. Consistently, TEMP is lower at 100 hPa (near the CPT) than at 146 hPa and its value is the lowest over the MariCont region. The IWC fraction is larger over MariCont than elsewhere in the tropies in the TL ( near 78%). While WV decreases by more than 8 ppmv in the TL over the MariCont region compared to the UT, the RHI in the TL reaches high values (RHI ~100 %) highlighting a saturated environment over the central South America, Africa, MariCont and the Western Pacific Ocean.~~

With the decrease of temperature from the UT to the TL (Figs. 2 and 3) and the associated decrease of WV at saturation, the observed WV decreases by more than 11 ppmv over the MariCont region and by around 5 ppmv in other regions. In the TL, the IWC and the fraction of water in the ice form is larger over MariCont (beyond 1 mg m-3 and near 78%, respectively) than elsewhere in the tropics. RHI in the TL reaches high values (RHI ~100 %) highlighting a saturated environment over the central South America, Africa, East of MariCont and the Western Pacific Ocean. In comparison to other tropical regions, the larger IWC over the MariCont can be explained by (i) the larger condensation of water vapour associated with the larger temperature drop from the UT to the TL, and (ii) a larger transport of ice into the TL by deep convection.

**9) line 192** 'To investigate the vertical distribution and the diurnal cycles of water species in the TL, we have defined seven tropical convective zones shown in Figure 4'
Comment: It would be better to introduce Fig. 4 earlier (see comment on line 46).

We introduced the MariCont, earlier in Fig.1 in the introduction. However, we decided to present the choice of the study zones after having shown the global tropical characteristics observed in Figs.1, 2 and 3. Thus, this is only from Figs. 1, 2 and 3 that we decided to selected the 7 study zones presented in Fig. 7.
We have changed line 315 in order to explain more how we have chosen these study zones:

'To investigate the vertical distribution and the diurnal cycles of water species in the TL, we have defined from results presented Fig.1–3, seven tropical convective zones shown in Figure 4:'

**10) line 211** 'RHI is lower in the UT than at the TL by ~10 % 'Comment: This is not a new finding – please provide references.

We have inserted a new reference Fueglistaler et al. (2009) and modified the selected sentence in line 338:

RHI is lower in the UT than at the TL by ~10 % (Fueglistaler et al., 2009) (Figs. 2d and 3d, respectively).

The reference is in the References section:
Fueglistaler, S., Dessler, A. E., Dunkerton, T. J., Folkins, I., Fu, Q. and Mote, P. W.: Tropical tropopause layer, Rev. Geophys., 47(1), RG1004, doi:10.1029/2008RG000267, 2009.

**11) line 213** 'The convective lifted ice does not sublimate as rapidly as in the UT ...'
Comment: in case of saturation or supersaturation, the ice does not sublimate at all ... as you state later in the paragraph, existing ice crystals even grow and sediment. I recommend to rephrase the sentence to provide a clear picture of the processes.

We changed the text line 338 by :

Since the UT is on average sub-saturated, convective-lifted ice can sublimate and can be seen as a source of WV. At the TL, as the atmosphere is in average close to saturation (RHI ~100%) over SouthAm, SouthAfr, MariCont and the Western Pacific Ocean (Fig. 3), the convective-lifted ice is statistically more often transported into a saturated or supersaturated region. In such context, the associated ice hydrometeors grow and sediment, and can also be transported down by convective downdrafts. These processes contribute to the loss of ice in this layer. Furthermore, the TL is the level of greatest dehydration because of supersaturation with respect to ice (Jensen et al., 1996; Jensen et al., 2005). In supersaturation conditions, the excess of water vapour can condensate on existing ice crystals (or form new ones in presence of condensation nucleus) allowing ice crystals to grow and sediment which dehydrates the layer. Conversely, hydration may occur in the LS (if reached by convection) because this layer is subsaturated with respect to ice (Khaykin et al., 2009; Allison et al., 2018; Dauhut et al, 2018).

By the way: dehydratation → dehydration (line 215)
We modified the error.

**12) Figure 5** The water vapor axis in panel b), I would recommend to use a logarithmic scale to make the differences in the TL better visible.

We changed the figure 5 b by the following new one with the logarithmic scale:

[Figure]

**13) line 237** 'Over land, all regions under consideration show a better efficiency to inject ice in the UT during Day than during N. is βx the IWC background.'
Comment: Something is wrong with the end of the sentence (marked in blue).

We modified the typo into line 367:

Over land, all regions under consideration show a better efficiency to inject ice in the UT during Day than during Night. $\beta_x$ is considered as the IWC background.

**14) line 259** 'Deep convection does not inject directly WV in the UT'.
Comment: Is that true ? Look for example the recently appeared paper:
Convective hydration in the tropical tropopause layer during the StratoClim aircraft campaign: Pathway of an observed hydration patch Keun-Ok Lee et al. https://www.atmos-chem-phys-discuss.net/acp-2018-1114/

The effect of deep convection on the WV in the UT and TL is primarily that the convection injects mainly ice, some of which sublimates to increase WV. This has been demonstrated in studies in Cloud Resolving Models, e.g. simulations of the Hector convective system to the north of Australia (Dauhut et al., 2015). In that case, the onset of the WV increase is delayed by 1-3 hours with respect to the onset of deep convection (~12:00 LT). For this reason, the space correlation between Prec and WV is very low compare to the space correlation between Prec and IWC (see results in section 3.3 and Table 1). For that reason, we now focus on the diurnal cycle of ice (rather than of WV) and on the amount of ice injected into the UT and TL traced by Prec.

However, as we say on the introduction, with some specific condition, some WV can enter directly into the LS by overshoots. Infact, WV injection into the TTL would be totally excluded only if the TTL is always supersaturated. For that reason, we changed the sentence line 259 by the new one line 391:

~~Deep convection does not inject directly WV in the UT. Modelling studies from Cloud Resolving Models based on the Hector convective system in North of Australia (Dauhut et al., 2015) show that WV in the UT and TL is produced by ice sublimation. The onset of the WV increase is delayed by 1-3 hours with respect to the onset of deep convection (~12:00 LT).~~

 $IWC_{Day,Night}^{146}$  $Prec_{Day, Night}$ ~~(R ~0.7) is larger than the space correlation $R$ betweenand(R ~0.2). For that reason, we now focus on the diurnal cycle of ice and on the amount of ice injected in the UT and the TL by deep convective systems traced by Prec.~~

Deep convection does not inject directly WV in the UT, but injects ice which then may sublimate into WV. Modelling studies with Cloud Resolving Models illustrate the sequence of mecanisms (ice uplift, mixing, sublimation) that leads convection to hydrate initially subsaturated layers (Dauhut et al., 2015, 2018, and Lee et al., 2018). In Dauhut et al. (2015), the onset of the WV increase is delayed by 1-3 hours with respect to the onset of deep convection (~12:00 LT). Such delay can explain why $WV_{Day, Night}^{146}$ and $Prec_{Day, Night}$ are not spatially correlated in observations ($R$ ~0.2), while the space correlation $R$ between $IWC_{Day,Night}^{146}$ and $Prec_{Day, Night}$ is stronger ($R$ ~0.7). For this reason, we now focus on the diurnal cycle of ice and on the amount of ice injected in the UT and the TL by deep convective systems traced by Prec.

**15) line 327** ' ... (the diagram over ocean would look different since maxima appear during local night ton).is defined as the time ...
Comment: Something is wrong with the end of the sentence (marked in blue).

We have changed the typo into line 458:

(the diagram over ocean would look different since maxima appear during local night). $t_{on}$ is defined as the time when Prec starts to...

**16) Figure 10** What is the dashed red line in panel b) ?

332 ' (see the period of proportionality in Fig. 10).' Is this the dashed red line ? Please indicate.

The red dashed line represents when $t_{on} \approx t_{on}^B$ (i. e. when the diurnal minimum of IWC occurs at the same time as the diurnal minimum of Prec). For more explanation about the red dashed line, we deleted one of the confuse back square and changed the caption of Fig.10 as follow:

Figure 2: Methodology to estimate the diurnal cycle of ice water content (IWC) over land: (a) diurnal cycle of precipitation, with representing the minimum of precipitation $\left(Prec_{min}\right)$ with its associated time $\left(t_{on}\right)$ and Prec during the day $\left(Prec_{Day}\right)$ at $t_{Day}$ = 13:30 LT. (b) Diurnal cycle of $IWC^B(t)$ in red solid line, estimated from the diurnal cycle of Prec and from the two MLS measurements of ice ( ◆ , $IWC_x^B$), with the timing of the onset of the ice injection at $t_{on}^B = t_{on} + \delta t^B$. ■ represents the IWC maximum $\left(IWC_{max}^B\right)$ and ▪ , the $IWC_{min}^B$ when $\delta t^B >$ 1h. Note that when $t_{on} \approx t_{on}^B$, the diurnal cycle of IWC is the red dashed line.

**17) line 333** 'This hypothesis considers deep convection represented by Prec as the main process bringing ice into the UT and the TL.'

Comment: Could you please further justify the hypothesis that deep convection is the main process bringing ice into the UT and the TL – what is with the thin cirrus that are not included in your analysis? They can also bring (or form) ice in these regions.

We modified the selected sentence (section 2). We explained that MLS is also able to measure thin cirrus. Thus, ice composing cirrus formed by deep convection during the growing phase of the convection are included in our calculation. However, our method is not able to make the difference between ice including cirrus formed by convection and cirrus formed by other processes.

Thus, we changed line 464:

This hypothesis considers deep convection represented by Prec as the main process bringing ice into the UT and the TL during the growing phase of the convection.

For more explanation we also changed the line 280:

 Deep convection can affect water vapour and ice in the UT and the TL whereas Prec measurements include the contribution of both shallow convection (that does not reach the UT) and deep convection (reaching the UT and the TL).

**18) line 332, 334** P(t) and Px : P means Prec ?

We corrected this typo. P = Prec.

19) line Figure 11 The colors in the panels are differently defined:

(a) red – IWCp , blue – IWP, black – Prec

(b) red – IWC , blue – PIWP, black – Prec

In the caption (and text) you define pIWP, IWC, Prec. Please clarify.

We corrected it.

**20) line 356** 4.2 Validation of the method with SMILES measurements

Comment:

**(a)** The SMILES instrument is important for the evaluation of the method, but it is not introduced (for example in Section 2) and no references are given. Please insert a brief description of the IWC detection method and the uncertainties of the measurements to convicingly demonstrate that SMILES is suitable to evaluate the data analysis method applied here.

We created a new session, l. 267, into the Section 2) to introduce SMILES and we added some references.

**2.3 SMILES**

The Superconducting Submillimeter-Wave Limb-Emission Sounder (SMILES) was a Japanese atmospheric limb sounding instrument on board the International Space Station (ISS) platform. SMILES measurements of ice are used in this study. The instrument measured IWC between 120 and 80 hPa and the measurement of tropospheric ice, the partial Ice Water Path (pIWP) integrated between 1000 and 180 hPa, during the short period of October 2009 to April 2010. As ISS follows a low Earth orbit, with about 15 orbits a day, the ice measurements in the troposphere and tropopause layer over the entire 40°N-40°S band can be provided in about two months. Thus, the austral convective season of DJF is enough to covers the entire tropical band. Furthermore, as each orbit is drifted about 20 min earlier each day, the entire diurnal cycle of ice can be provided during this period. In the present study, SMILES measurements will be used as reference of diurnal cycle of ice and will be compared to the climatology of the diurnal cycle of ice estimated in the UT and the TL.

References about previous SMILES studies are presented later in the section 4.2, and in the references:

The diurnal cycle of IWC and pIWP calculated from SMILES has shown the well separated signal over tropical land and ocean (Millán et al., 2013; Jiang et al., 2014).

Jiang, J. H., Su, H., Zhai, C., Janice Shen, T., Wu, T., Zhang, J., et al.: Evaluating the Diurnal Cycle of Upper-Tropospheric Ice Clouds in Climate Models Using SMILES Observations. J. Atmos. Sci. 72, 1022–1044., doi:10.1175/JAS-D-14-0124.1, 2014.

Millán, L., Read, W., Kasai, Y., Lambert, A., Livesey, N., Mendrok, J., Sagawa, H., Sano, T., Shiotani, M. and Wu, D. L.: SMILES ice cloud products, J. Geophys. Res. Atmospheres, 118(12), 6468–6477, doi:10.1002/jgrd.50322, 2013.

**(b)** Validation : I suggest to better use 'Evaluation' , since also the SMILES measurements are no absolute proof.

We changed « Validation » by « Evaluation » in the title of the session. l. 486.

**21) line 393** 'NOAA Interpolated Outgoing Longwave Radiation (OLR) during DJF 2009-2010 is consistent with the fact that the convective activity grows higher over land than over ocean (not shown).'
Comment: What do you mean with 'consistent'? Even if a figure is not presented, the ranges of OLR over land and over ocean should be specified.

We clarified (l. 523) the selected sentence into:

NOAA Interpolated Outgoing Longwave Radiation (OLR) during DJF 2009-2010 is lower over land (mean OLR between 195 and 215 W m$^{-2}$) than over ocean (between 235 and 270 W m$^{-2}$) which is consistent with the more intense land convective activity compared to that of oceans.

**22) line 417**
how = show
modified

**23) line 423** 'Table 4 presents ton TL, Δt TL, IWCmin TL and ΔIWC TL...'

Comment: ton TL is not shown in Table 4.

    We modified the wrong sentence in the caption of Table 4:

        **Table 4: As a function of the delay ($\delta t$ = 0, 1, 2, or 3 h) between the beginning of the Prec onset and the IWC onset in the tropical tropopause layer (TL):**  $(\dot{c} t > \dot{c}_{on}^{TL})$, $\dot{c}$ **duration of the injection of ice in the TL $\left( \Delta t^{TL} \right)$, minimum amount of ice in the TL $\left( IWC_{min}^{TL} \right)$ and amount of ice injected into the TL $\left( \Delta IWC^{TL} \right)$ as a function of the 6 study zones and averaged over the land and ocean zones ($\mu$(Lands ZonesTropics), $\mu$(Oceans ZonesTropics), respectively) during DJF from 2004 to 2017. The bolded values highlight the most important $IWC_{min}^{TL}$ and $\Delta IWC^{TL}$ and the associated regions.**

**24) line 440** '... all these results are consistent with the OLR showing strongest values over region having highest OLR signal (not shown).' ???

    We decided to remove this confuse sentence line 572:

**25) line 447** '... convective processes during the dissipating stage,...'

Comment: As long as convection is active, how can the ice clouds are in a dissipatig stage ? Or do you mean '... decreasing convective processes ...' ? See also line 452

    'Dissipating stage' is a term used to describe the period during which the convection is decreasing (Takahashi and Luo, 2014). During this stage, downdraft processes are more important than updraft processes. For instance, we present Figure R1 the definition of the convective dissipating stage. (During the dissipating stage, the ascending convective transport weakens or disappears and the other processes such as sublimation, sedimentation and large-scale transport and mixing then act to decrease the local ice concentrations.)

[Figure]

▲ **Cumulus stage** Vertical motions are limited by mixing with cool, dry environmental air aloft. But the mixing adds water droplets that evaporate and cool the surrounding air, creating instability.

▲ **Mature stage** In this stage, there are well-organized updrafts and downdrafts. Updrafts, which can reach as high as the tropopause, spread out to form an anvil cloud. Downdrafts are created by falling precipitation and entrainment of cooler, drier environmental air.

▲ **Dissipating stage** Dissipation occurs when cool, dry environmental air mixes into the cloud, inhibiting convection and latent heat release.

**4.28 Stages in the development of an air-mass thunderstorm**
The three development stages of an air-mass thunderstorm are the cumulus, mature, and dissipating stages. Each stage has characteristic vertical winds and precipitation.

Figure R1. Illustration and definition of the cumulus stage, mature stage and dissipating stage of an air-mass thunderstom from http://geography.name/thunderstorms-2/.

To be more clear, we added line 122:

Finally, the influence of the downdraft convective dissipating stage on the decreasing phase of the IWC diurnal cycle in the UT and the TL is discussed in Section 5 and conclusion will be drawn in Section 6.

and line 584:

This loss can be caused by several processes including sedimentation, downdraft convective processes during the dissipating stage, sublimation into water vapour or horizontal advection and mixing.

**26) line 483** 'Other processes may play a minor role such as the decrease of the temperature in the TTL, increasing the saturation ratio and allowing the crystal nucleation and growth, or the bring of ice in the UT and TL by horizontal advection.'

Comment: How do you know that these processes are minor? Do you know their order of magnitude in comparison to deep convection? From my feeling your conclusion is right, but for an ensured statement I think an estimate of the other processes is needed.

Firstly, we changed in the text:

Other processes may play a minor role such as the decrease of the temperature in the TTL, increasing the saturation ratio and allowing the crystal nucleation and growth, or the transport of ice in the UT and TL by horizontal advection.

Then, to answer to your questions and to be more clear in the text, we added the following sentences:

The decreasing phase of the ice diurnal cycle is also evaluated with Prec and discussed. While, it is difficult to quantify the impact of other processes than convective processes on the diurnal cycle of ice in the UT and the TL, we are able to assess that the deep convection impacts on the depletion of ice during the decreasing phase of the ice diurnal cycle (from Table 5: 97 % in the UT over the MariCont and ~80 % in averaged UT and TL over other regions). Thus, Prec is also considered to be a good proxy of the decreasing phase of the convection, especially over the MariCont-L region in the UT (to within 3 %, according to Table 5), and convective processes are the main processes impacting the decreasing phase of the diurnal cycle of ice. Furthermore, our study has shown that the estimated diurnal variation of ice is the largest in the regions (identified from TRMM) where the diurnal variation in temperature is also the largest although horizontal transport may play a role, but it cannot be quantified using our methodology.

**27) line 491** '.. the injection of ice over the MariCont-L into the UT and the TL ... is the greatest in the tropics.'

Comment: So your finding is that deep convection is the major process feeding the 'stratospheric fountain' and not large- scale three-dimensional circulation, yes ? You mentioned in the abstract that the relative importance of the two processes continues to be debated – so I think you could better pronounce the importance of your finding. This is true also for line 495

Our study is mainly focus on the UT and TL layers and does not conclude on the water or ice entry into the LS. Water vapour entering the stratosphere may be determined by large-scale temperatures or perhaps by overshooting convection. However, results in our study could provide some new information to other studies about the water entrance into the LS via deep convection. For that reason, we decided to change the sentence l 623:

It has been shown that the injection of ice over the MariCont-L into the UT and the TL ($\Delta IWC^{UT}$ = 3.34 mg m$^{-3}$, $\Delta IWC^{TL}$ = 0.56-0.42 mg m$^{-3}$) is the greatest in the tropics. This

injection of ice could have a strong importance on the amount of ice entering into the LS and feeding what has been called the 'stratospheric fountain' (Newell and Gould-Stewart, 1981) observed over the MarCont region.

We also changed the sentence line 629:

In summary, while the importance of deep convective processes and large-scale three-dimensional circulation processes on the ice and water injection up to the UT, TL and LS are still debated, our study shows that the ice diurnal cycle in the UT and the TL is mainly governed by vertical processes linked to the convective activity that are much stronger than other processes such as e.g. horizontal mixing, sublimation, sedimentation, ...

Two papers that might be of interest for you study:
Robrecht et al. (2018): Mechanism of ozone loss under enhanced water vapour conditions in the mid-latitude lower stratosphere in summer , ACPD.
Smith et al. (2017):
A Case Study of Convectively Sourced Water Vapor Observed in the Overworld Stratosphere over the United States, JGR.

Thank you. However we do not use these two papers in the references because, these are both about water vapour injection into the stratosphere and its consequences (in midlatitudes), while we are focusing on the UT and the TL layers only.

**RESPONSES TO THE ANONYMOUS REFEREE #2**

**General remarks.**

**1)*** The introduction places the study into the context of stratospheric water vapour and related problematics. With that, in my opinion, the obtained results are of limited relevance for the control of stratospheric water budget. The stratospheric entry of water is mostly driven by minimum temperature at the Cold Point Tropopause and, to a much smaller extent, by injection of ice into subsaturated environment above this level. The CPT level corresponds to 82 hPa level of MLS (as can be clearly seen in Fig. 5a), whereas the analysis is performed for 146 and 100 hPa levels. The ice water detrained below CPT may have very little or no impact on stratospheric water: even if the injected ice crystals sublimate before settling down, the amount of water vapour ascending into the stratosphere would be limited by the colder temperatures at CPT level. Moreover, large-scale convection may lead to additional cooling of CPT at diurnal and intra-diurnal time scales thereby further limiting the cross-tropopause transport of water. I believe the above considerations should be discussed in the context of the role of deep convection of stratospheric water. It would also be useful to compare the results regarding IWC obtained for 100 hPa with those for 82 hPa level.

We modified the whole introduction in order to move the focus away from stratospheric water vapour (Referee #2) (and to provide more justification for the work that is presented (Referee #1)).

**2)*** On the base of comparison between TRMM Prec and TRMM OPF diurnal cycles, Prec is shown to be a good proxy of deep convection during its growing phase. Could you clarify why the OPF data as such could not be used for the IWC analysis?
Indeed, OPF would be a better proxy of deep convection reaching the TL than Prec. However, the OPF fields were only available for the period studied in Liu and Zipser (2005),

namely from January 1998 to November 2000 and from December 2001 to December 2003, and were not accessible to the scientific community. Consequently, we ought to find another proxy covering the 2004-2017 period. However, it could be very interesting to find other instruments available during 2004-2017, able to built the same estimation than the OPFs (originally calculated from the combination of TRMM observations and different instruments detailed in Liu and Zipser (2005)).

In our study, Prec has been measured by TRMM over the same period than IWC from MLS: 2004-2017. In addition to the comparison of Prec and IWC, we also show that Prec behaves as OPF during the growing phase of the deep convection in austral convective seasons of DJF.

We have inserted a new paragraph l. 438:

> In summary, although OPFs are representative of deep convection reaching the TL, OPFs developed and presented in Liu and Zipser (2005) are not available during our study period of 2004-2017. However, our results in this section have shown that during the growing stage of deep convection, Prec is a good proxy of convection reaching the TL over land and ocean. Thus, as Prec is available in time coincidence with MLS IWC ans WV over 2004-2017, we will use Prec to interpret the time evolution of IWC in the UT and the TL.

**Specific remarks.**

**3) l. 18-19.** The purpose of the method is missing in this sentence, i.e. "We propose a method for..using..."

We modified the specified sentence line 20 into:

> We propose a method that uses Prec as a proxy of deep convection bringing ice up to the UT and TL during the growing stage of the convection, in order to estimate the amount of ice injected in the UT and the TL, respectively.

**4) l. 165.** The title of Sect. 3.1 should be more specific.

We changed it line 284 into:

> Tropical distribution of Prec and water budget in the UT and TL

**5) l. 180.** The statement regarding a ubiquitous subsaturation in the Austral convective regions is surprising. Consider that the RH product of MLS may have a substantial dry bias at 82 hPa since it is based on MLS temperature profiles. The latter, in turn, do not resolve the sharp temperature minimum at CPT, which leads to underestimation of RHI at this level. While the geographical distribution of MLS RH could be fairly accurate, the quantitative statements based on these data should be avoided. A correct way to infer RHI values from MLS measurements would be to compute them from WV profiles of MLS and temperature profiles from a reanalysis data set with sufficient vertical resolution in the TTL, e.g. MERRA-2.

MLS provides RHI measurements with a vertical resolution of about 5 km, which . The validation of the RHI product from MLS is presented in Read et al. (2007). More precisely, from 316–0.002 hPa, RHI is derived from the standard products of water and temperature using the Go-Gratch ice humidity saturation formula. Furthermore, our analysis is climatological, thus RHI is a climatological value. It may be possible that RHI > 100 % on some specific locations and periods.

Our results presented in Fig. 2 show that RHI is below 100% in the UT (146 hPa) whereas RHI is reaching 100% in the TL (100 hPa). Thus, we modified the sentence l. 297 into:

> The RHI is higher over land than over ocean but RHI never exceeds 100% at 146 hPa, on average at 2°x2° resolution (consistently with the RHI tropical vertical profil shown by Fueglistaler et al., 2009). The MariCont region exhibits the highest RHI with values close to 100%.

**6) l.203.** Since Fig. 5a shows the profiles at the native MLS pressure levels, a correct inference on the CPT pressure would be 83 hPa ±half width of MLS weighting function.

With the information given by MLS on temperature observations, we are not able to give the pressure level of the CPT to better than the half-width at the maximum of the associated averaging kernel. In Fig. 5a, we expect the CPT to be between 100 and 82 hPa that is already widely known. In order to support these statements, we decided to add 2 references: Fueglistaler et al. (2009) (their Figure 5 shows temperatures at different pressure levels) and Kim and Son (2012) (their Figure 3 shows CPT temperatures).

We modified in section 3.2, line 335, the following sentence:

> In Fig. 5a, the tropical CPT is found in the pressure range of 82 hPa ± half width of the associated averaging kernel. Figure 6 presents the tropical tropopause pressure defined by NCEP and shows an averaged pressure (~ 100-115 hPa) closed to the pressure given by MLS at 100 hPa, named the TL in this study. Based on previous studies (Fueglistaler et al., 2009 and Kim and Son, 2012) and fig 6, we can nonetheless narrow the TTL range to 100-80 hPa.

We added Kim and Son (2012) into the References section line 701:

> Kim, J. and Son, S.-W. Tropical Cold-Point Tropopause: Climatology, Seasonal Cycle, and Intraseasonal Variability Derived from COSMIC GPS Radio Occultation Measurements. J. Climate, 25, 5343–5360, doi: 10.1175/JCLI-D-11-00554.1, 2012.

Furthermore, Fig. 5a indicates which region has the lowest/highest temperature profiles. In order to complete this Figure, we choose to present Fig. 6 which gives the pressure level of the tropopause.

**7) l.204-207** and Fig. 6. What is the relevance of this information in the context of Sect.3.2 entitled "Water budget in the UT and TL"? Please clarify.

In Fig. 6, we found the tropopause between 100 and 115 hPa what is close to the layer that we called "TL", defined at 100 hPa. Fig. 6 shows the longitudinal pressure differences of the Tropopause and completes Fig. 5. From Fig. 6, the Tropopause is found to be **the lowest** over the MariCont whereas, from Fig. 5a, the tropopause is found to be **the coldest** over the MariCont.

Furthermore, from Fig. 6, the tropopause is found to be higher over ocean than over land. Thus, these two figures provide complementary information on the tropical tropopause layer area. From Fig. 5a, the CPT is near 82 hPa for all study zones, whereas the tropopause defined by NCEP is close to 100 hPa.

We have inserted a new paragraph in this section, l. 327:

> To investigate the processes in the UT and TL which drive the water budget, the vertical profiles of TEMP, WV, IWC and RHI are shown for the different study zones in Figure 5. To complete the comparison between the study zones, the DJF average of the tropopause pressure level is represented over all the tropics in Figure 6 from the National Centers For Environmental Prediction (NCEP).
> In Fig. 5a, the tropical CPT is found between 100 and 80 hPa consistently with previous results presented in e.g. Fueglistaler et al. (2009) and Kim and Son (2012). The tropical tropopause defined by NCEP (Fig. 6) is close to the level given by MLS at 100 hPa we named the TL in this study (~ 100-115 hPa). Furthermore, the tropopause pressure shown in Fig.6 is lower (higher altitude) over ocean than over land. While the pressure at the tropopause over West Pacific, South Arabia, Caribbean sea present is the lowest (~105

hPa). Furthermore, the tropopause is the coldest (Fig. 5a) and has the highest pressure (lower altitude) (Fig. 6) over the MariCont.

**8) l.282-283.** The section title refers to UT level, whereas the first line reads " ...convection reaching the tropopause … "

We modified the specified sentence line 143 into:

l. 298 :

In this section, we quantify the link between deep convection reaching the UT and Prec.

**9) Sect. 4.2.** For consistency, the description of SMILES instrument should be introduced in Sect. 2. Please clarify how the full diurnal coverage is ensured with ISS platform.

This point has been detailed in the response to the referee #1 (point 20)

**10) Figure 1.** Color scales of 1a and 1b should be the same. Figure 1c is a duplicate of 1a. Technical corrections.

This point has been detailed in the response to the referee #1 (point 5).

**11) l.175.** Maximal => maximum
Done.

**12) l.213.** Convectively-lifted?
Done

**13) l.221.** Spatial correlation
Done

**14) l.223.** Consider rephrasing the sentence ending with "… WV and IWC in the UT and Prec is analyzed".

l.238, we modified the specified sentence line 351 into:

In order to quantify the relationship between deep convection and the budget of water injected in the UT, the correlation between both WV and IWC in the UT and Prec is analysed.
This section presents the relationship between deep convection and the water vapour and ice injected in the UT.

**15) l.237-238.** Broken sentence

See point #13 of the Referee #1.

**16) l.376.** try "before reaching"

Done.

**17) l.417** sentence unclear

l.433, we changed :

[revised manuscript text omitted]

---

## Author Response (AR2)

**Manuscript Title: Ice injected up to the Tropopause by Deep Convection: 1) in the Austral Convective Tropics by Dion et al.**

We thank the two referees as well as the Co-Editor for this decision to publish the manuscript subject to minor revisions. We adress the following answer to the minor revisions and we hope we have addressed appropriately the issues raised by the co-editor.

The co-editor's comments are repeated in the following paragraphs in blue and our responses appear in red. Copy of sentences or words changed into the paper are repeated and appear in yellow. Sentences or words deleted into the paper appear crossed-out and in orange.

**RESPONSES TO THE CO-EDITOR :**

**Co-Editor Decision: Publish subject to minor revisions (review by editor)** (18 Mar 2019) by Rolf Müller

Comments to the Author:

Dear Authors,

based on your revised version and based on the reviewers comments, I am happy to accept your paper subject to minor revisions. There is one remaining comment by referee #2 which I would ask you to consider in your revised/final version of your paper (it was only contained in the comments to the editor, not in the comments to the authors).

One detail: l. 23: "diurnal amount" is unclear. Do you mean " diurnal cycle of injection"?

Yes indeed, we changed the expression according to your suggestion line 23:

> Next, the diurnal  cycle of injection of IWC  into the UT and the TL by deep convection is calculated by the difference between the maximum and the minimum in the estimated diurnal cycle of IWC in these layers and over selected convective zones.

Greetings

Rolf

Original comment:

4) line 127 'The MLS IWC sensitivity thresholds do not allow to detect low ice content such as cirrus outflow associated with convective events. MLS IWC sensitivity will mostly be able to detect ice from convective cores. Following Livesey et al. (2017), we will not consider IWC measurements less than 0.02 mg m−3.'

Comment: All the thin cirrus are missed. How does that impact the results ?

Answer:

This sentence was not clear enough and have been changed. The detection limit of MariCont ice concentration is 0.02 mg/m-3 in the TL whatever the origin/source of ice: convective cores or cirrus associated with convective events. Thus, cirrus are not missed.

The new sentence l. 127 is now l.135:

The MLS IWC sensitivity thresholds are 0.1 mg

m −3 in the UT and 0.02 mg m −3 in the TL (Livesey et al., 2017) and will imply a small

under-estimation of the IWC measured values.
Furthermore, we changed according the detection limits values in the Tables
2 and 3.

New comment: The IWC in the UT and TL can go down to $10^{-3}$ -$10^{-4}$ mg/m3 (see Krämer et al., 2016, ACP, their Figure 2). So 'small under-estimation' is formulated too modest, since many cirrus are missed.

We deleted one word in the incriminated sentence (line 144):

The MLS IWC sensitivity thresholds are 0.1 mg m$^{-3}$ in the UT and 0.02 mg m$^{-3}$ in the TL (Livesey et al., 2017) and will imply an  under-estimation of the IWC measured values.